# DreamSim: Learning New Dimensions of Human Visual Similarity using Synthetic Data

**Stephanie Fu**[*1]    **Netanel Y. Tamir**[*2]    **Shobhita Sundaram**[*1]

**Lucy Chai**[1]    **Richard Zhang**[3]    **Tali Dekel**[2]    **Phillip Isola**[1]

[1]MIT    [2]Weizmann Institute of Science    [3]Adobe Research

Figure 1: **What makes two images look similar?** We generate a new benchmark of synthetic image triplets that span a wide range of mid-level variations and gather human judgments, rating whether image A/B is more similar to the reference. Our benchmark spans various notions of similarity such as pose (top-left), perspective (top-mid), foreground color (mid-left), number of items (mid-right), and object shape (bottom-left). This allows us to learn a new metric (DreamSim) that better coincides with human judgments w.r.t. existing similarity metrics (LPIPS) or embedding-based metrics extracted from recent large vision models (DINO & CLIP).

## Abstract

Current perceptual similarity metrics operate at the level of pixels and patches. These metrics compare images in terms of their *low-level* colors and textures, but fail to capture mid-level similarities and differences in image layout, object pose, and semantic content. In this paper, we develop a perceptual metric that assesses images holistically. Our first step is to collect a new dataset of human similarity judgments over image pairs that are alike in diverse ways. Critical to this dataset is that judgments are nearly automatic and shared by all observers. To achieve this we use recent text-to-image models to create synthetic pairs that are perturbed along various dimensions. We observe that popular perceptual metrics fall short of explaining our new data, and we introduce a new metric, *DreamSim*, tuned to better align with human perception. We analyze how our metric is affected by different visual attributes, and find that it focuses heavily on foreground objects and semantic content while also being sensitive to color and layout. Notably, despite being trained on synthetic data, our metric generalizes to real images, giving strong results on retrieval and reconstruction tasks. Furthermore, our metric outperforms both prior learned metrics and recent large vision models on these tasks. Our project page: https://dreamsim-nights.github.io/

* Equal contribution, corresponding authors. Order decided by random seed.

# 1   Introduction

*"A sense of sameness is the very keel and backbone of our thinking" – William James, 1890*

Our understanding of the visual world hinges crucially on our ability to perceive the similarities between different images. Moreover, humans can reason about many notions of similarity, ranging from low-level perceptual properties such as color and texture, to higher-level concepts such as an object's category or a scene's emotional valence [76]. This capacity to conduct meaningful visual comparisons underlies our ability to effortlessly transfer our knowledge to new environments, e.g., recognizing unseen or unfamiliar objects based on their relatedness to familiar ones [42, 70] .

Computer vision has tried to capture this sense of similarity with low-level metrics like PSNR and SSIM [88], as well as learned perceptual metrics such as LPIPS [94] and DISTS [24]. Despite their utility, these metrics are limited in that they focus on the pixel or patch level and fail to capture higher-level structures. These limitations have motivated researchers to reach for *image-level* embeddings from large vision models such as DINO or CLIP to measure image-to-image distances in a large variety of applications [12, 40, 41, 44, 62]. Recent studies have shown that these embeddings do well at capturing certain high-level similarity judgments, in particular, predicting which semantic categories will be considered alike by humans [59]. It remains unclear, however, how well these models align with human perception of richer and more fine-grained visual structure.

In this paper, we introduce a new perceptual metric, which bridges the gap between lower-level patch-based metrics and broad categorical comparisons. We collect a new dataset named NIGHTS – Novel Image Generations with Human-Tested Similarity – containing human similarity judgments over image triplets. Each triplet consists of a reference image and two perturbed versions, along with human judgments as to which version is most similar to the reference (Fig. 1). We use iterative filtering together with recent diffusion models to collect our dataset, which is designed to capture image sets that are cognitively impenetrable (i.e. result in consistent decisions across different individuals) yet showcase rich variations. For example, our data contains images of similar object appearance, viewing angles, camera poses, overall layout, etc. This dataset differs qualitatively from prior low-level datasets [94], which focused on perturbations like blurring and adding noise, and from previous high-level datasets [34, 35], which showed variation just at the level of categories (e.g. "is an image of a kitchen more like an image of a giraffe or an image of a beach").

On the mid-level similarity task presented by our data, we find that features from recent large pre-trained vision models [10, 39, 66] outperform the current set of standard perceptual metrics [24, 59, 94]. We further show that these large vision models can be tuned on our data to be substantially more human-aligned. Our resulting metric, DreamSim, can be dropped into existing pipelines and demonstrates high agreement with human visual perception in both quantitative assessments and qualitative comparisons using out-of-domain real images (e.g., image retrieval, image synthesis). We also analyze which features our metric is most sensitive to and find that, compared to previous perceptual metrics, it focuses relatively heavily on foreground objects, while compared to modern image embeddings, it does not neglect color and layout.

In summary, our contributions are the following:

- A new image similarity dataset, consisting of 20k synthetic image triplets designed to be cognitively impenetrable with human judgments as labels.
- A tuned metric that captures how humans naturally perceive image similarity, achieving 96.16% accuracy in predicting human judgments on our dataset.
- Analysis of the image properties that affect our model's decisions.
- Demonstration of downstream applications to image retrieval and synthesis.

# 2   Related work

**Perceptual similarity.** Classical metrics such as Manhattan $\ell_1$, Euclidean $\ell_2$, MSE, and PSNR use point-wise difference to measure similarity, thus failing to capture important joint image statistics. Patch-based metrics, including SSIM [88], FSIM [92], and HDR-VDP-2 [55] tackle this issue and are widely used in applications involving photometric distortions such as image quality assessment. However, they do not capture the nuances of human vision when more structural ambiguity is present [73] and are not suited for more complex image generation tasks.

With the deep learning revolution, classical metrics have been replaced by learning-based metrics [26, 29, 43]. These metrics are defined in the space of deep features extracted from pre-trained networks, such as VGG [78] or AlexNet [47]. Amir and Weiss [4] demonstrate that even *untrained* networks can be adapted as perceptual metrics. Zhang *et al*. [94] observe that feature-based metrics outperform classical metrics across different convolutional architectures and learning paradigms, suggesting that perceptual similarity is an emergent property in deep representations. Further tuning on the perceptual data yields improvements, such as in LPIPS [94], PIE-APP [65], or DPAM in the audio domain [53, 54]. Further improvements include ensembling for robustness [45], antialiasing for stability [24, 36, 93], and global descriptors for texture [24]. Muttenthaler *et al*.[59] provide insight into *high-level* human similarity by training on a subset of the THINGS [35] dataset, focusing on concept similarity and omitting visual cues for images within a category.

While strong computer vision features make for strong perceptual metrics, counterintuitively, they eventually become *decorrelated* with perceptual similarity [20, 48]. Today, the predominantly-used perceptual metric is LPIPS, operating on $64\times64$ patches.

**Recent foundation models as metrics.** Foundation models provide strong pre-trained backbones for a variety of downstream tasks. These models primarily leverage the Vision Transformer (ViT) [25] architecture and are trained through self-supervised learning [16, 32, 81]. CLIP [66] learns to map images and text captions into a shared embedding space, proving useful for many (often zero-shot) tasks [41, 67, 90]. CLIP has been employed as a perceptual metric to train models for semantic consistency [12, 87]. Another self-supervised ViT-based model, DINO [10], extracts disentangled appearance and structure descriptors that can be employed in image generation pipelines. Amir *et al*.[5] show that DINO encodes valuable semantic information about object parts. Our work aims to systematically evaluate such representations for perceptual similarity, also including OpenCLIP (an open-source implementation of CLIP) [39] and pre-trained masked autoencoders (MAE) [31].

**Perceptual tests.** The two alternative forced choice (2AFC) test has historically been used by behavioral psychologists to study decision-making [51, 56]. Humans judge the similarity between two images by choosing to consider certain dimensions of similarity more than others [85]. Gathering judgments on ambiguous sets of images can be cognitively penetrable, calling upon a subject's cognitive processes rather than a more automatic, "wired-in" sense that is stable across humans and over time [11, 19, 79]. Previous studies have raised concerns about cognitive penetrability [57, 94]. On the other hand, as a psychophysical measure, just noticeable difference experiments (JND) are thought to be independent of subjective biases [3]. We follow best practices [94] and collect judgments on both of these complementary perceptual tests.

**Synthetic data.** GANs [30] have been used widely for dataset generation on tasks such as visual alignment [63], face manipulation [86], and adversarial training for image synthesis [77]. In recent years, text-driven generative models (e.g., Stable Diffusion [71], Imagen [72], DALLE-2 [68], MUSE [13]) have emerged as powerful tools for image synthesis. They have also been used to generate training data for a variety of downstream tasks [6, 9, 33, 75].

## 3  Perceptual dataset collection

While previous datasets focus on *low-level*, patch-based distortions [2, 94] or *high-level*, categorical [35] changes, we aim to close the gap, capturing distortions including mid-level variations. We aim to produce images with an underlying semantic commonality, but variations in a diversity of factors, such as style, color, pose, and other details, so that a human can assess their visual relationship. In Section 3.1, we describe our data generation pipeline – we prompt Stable Diffusion for related images of a given category, leveraging its natural image prior for variations within the category. We then describe our mechanism for collecting cognitively impenetrable perceptual judgments in Section 3.2.

### 3.1   Generating images with varied distortions

We leverage Stable Diffusion v1.4  [71], which generates diverse and high-quality images that adhere to a given text prompt. We sample images with a prompt of the same category, using the structure ``An image of a <category>''. The <category> is drawn from image labels in popular datasets: ImageNet [21], CIFAR-10 [46], CIFAR-100 [46], Oxford 102 Flower [60], Food-101 [8],

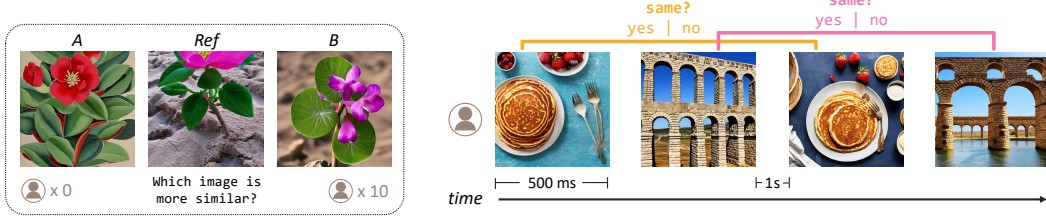

Figure 2: **Overview of 2AFC and JND data collection.** (Left) We collect up to 10 similarity judgments for each triplet in our dataset, and filter for unanimous examples. (Right) We validate our 2AFC results with a JND study. We flash two interleaved image pairs, and ask if the first & third, and second & fourth were the same. Interleaving the pairs ensures that users decide solely based on their initial reaction to each image from a standardized (500ms) viewing time.

| Dataset | Perturbation Type | Data Source | Input type | # Images / Patches | # Samples | # Judge per sample | Judgment type | Examples |
|---|---|---|---|---|---|---|---|---|
| BAPPS [94] | low-level (traditional & CNN-based) | MIT-Adobe5k, RAISE1k | 64×64 patches | 563.1k | 187.7k | 2.6 | 2AFC | |
| | | | | 19.2k | 9.6k | 3.0 | JND | |
| THINGS [34] | high-level conceptual | Google, Bing, Ebay, Flickr scrape | Images | 1.8k | 4.7M | 1 | Odd-one-out | |
| NIGHTS (Ours) | low & mid-level synthetic | Diffusion-synthesized | Images | 60.0k | 20.0k | 7.1 | 2AFC | |
| | | | | 411 | 137 | 3.0 | JND | |

Table 1: **Perceptual dataset comparison.** While previous work targets low-level and high-level variations, we leverage *synthetic* data to generate diverse perturbations. In addition, while previous data gathers fewer judgments per sample, we filter for unanimous judgments, curating cleaner samples.

and SUN397 [91]. While the ImageNet labels make up $\sim 55\%$ of the categories, the superset of categories across these datasets encompass a wide range of objects and scenes.

As illustrated in Figure 1, the resulting images display mid-level variations, such as pose, perspective, and shape, falling in-between pixel/patch-based distortions and categorical differences. Additional triplets and the full list of categories are in the Supplementary Materials (SM). We generate an initial set of 100,000 image triplets, which are later filtered during the labeling process. Each triplet consists of a reference image and two "distortions", generated from the same prompt.

## 3.2 Human perceptual judgments

We strive to gather perceptual judgments that are *automatic* (requiring little to no cognition), *stable* (invariant to changes in mental representation), and *shared* across humans, which requires a cognitively impenetrable judging process. Our main dataset consists of two alternative forced choice (2AFC) judgments on a triplet. Additionally, we collect just noticeable difference (JND) judgments on image pairs, corroborating our findings. Table 1 shows an overview of our dataset, compared to prior perceptual datasets.

**Two alternative forced choice (2AFC).** Figure 2 (left) illustrates our data gathering setup. We collect judgments on the commonly-used Amazon Mechanical Turk (AMT) platform [12, 83, 94]. We show participants triplets of images $(x, \tilde{x}_0, \tilde{x}_1)$ and ask whether $\tilde{x}_0$ or $\tilde{x}_1$ ("distortions" of $x$) is more similar to the reference image $x$. We collect judgments $y \in \{0, 1\}$ denoting the more perceptually similar distortion. This choice does not necessarily need to be easily articulated but should be cognitively impenetrable, that is, instinctive and consistent across humans.

Additionally, we use sentinels to ensure high-quality responses. Sentinels are designed as $(x, x, v)$, where one "distortion" is the image itself and the other is from a completely different prompt. Approximately $20\%$ of participants failed at least one sentinel, and we discard all their responses.

To collect cognitively impenetrable triplets, we design our 2AFC experiments to ensure that each triplet included in the dataset obtains a unanimous vote for either $\tilde{x}_0$ or $\tilde{x}_1$. We divide our experiments into 10 rounds, starting with 100,000 triplets, advancing instances that maintain unanimous votes. Triplets retained through all 10 rounds can earn a maximum of 10 votes, but may have less as a result of the aforementioned sentinels. To maintain dataset quality, we only include triplets with $\geq 6$ unanimous judgments in the final dataset.

Ultimately, our 2AFC dataset $\mathcal{D}^{2\text{afc}} = \{(x, \tilde{x}_0, \tilde{x}_1), y\}$ consists of 20,019 triplets with an average of 7 unanimous votes each. We partition our resulting dataset into train, validation, and test components with a random 80/10/10 split. Our dataset is publicly available on our project page.

**Just noticeable differences (JND).** JND aims to characterize the boundary when a distortion becomes *just* noticeable. Below a threshold, a small perturbation (e.g., a 1-pixel shift) appears identical to a human observer. This perceptual test provides a complementary signal for perceptual similarity and allows us to exploit uniform timing and the presence of a correct answer.

Intuitively, an image pair that falls below the JND threshold should be more likely to be selected as similar in the 2AFC test; thus a high correlation between 2AFC and JND scores indicates that our judgments are reliable under multiple experimental settings. To assess how well our perceptual tests agree, we collect judgments on the triplets from the 2AFC test set, filtering for triplets that "straddle" the JND threshold. Specifically, given triplet $(x, \tilde{x}_0, \tilde{x}_1) \in \mathcal{D}^{2\text{afc}}$, we independently collect JND judgments on pairs $(x, \tilde{x}_0)$ and $(x, \tilde{x}_1)$, keeping triplets where humans find *one and only one* to be identical. In total, our JND dataset contains 411 triplets $\mathcal{D}^{\text{jnd}} = \{(x, \tilde{x}_0, \tilde{x}_1), s\}$, where users chose one of $s \in \{0, 1\}$ to be identical.

We show our JND procedure in Figure 2. We interleave two images and their respective distortions into sequence $x \to v \to \tilde{x} \to \tilde{v}$, and ask whether the images in each pair were identical. This exploits masking, a perceptual phenomenon that occurs when an image is obscured by another. By masking with a time gap, we standardize viewing time across users and prompt for a decision based on an initial reaction and memory of the image. For each user, we show 48 distorted pairs, along with 24 identical pairs, to balance the expected responses. We collect 3 judgments per pair, with no person seeing a repeat, and take the majority vote as the label. Users mark 20.4% of distorted images being identical, indicating that some of our distortions indeed fall below the noticeable threshold.

# 4 Perceptual metric learning

We next evaluate how well pretrained embeddings and learned metrics align with our data, and we investigate if alignment can be improved by fine-tuning.

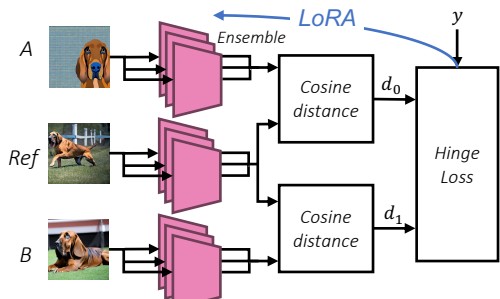

Figure 3: **Training overview.** Our perceptual metric is an ensemble of backbones, concatenating the representations together and fine-tuning with LoRA. We evaluate how well cosine distance on candidate feature spaces aligns with perceptual judgments $y$.

## 4.1 Embeddings as a distance metric

We denote a distance between two images as $D(\cdot, \cdot; f_\theta)$, where $f_\theta$ is a feature extractor. We evaluate a variety of state-of-the-art candidate embeddings and metrics. LPIPS [94] and DISTS [24] use CNN backbones, whereas DINO [10], CLIP [66], OpenCLIP [17], and MAE [31] use transformer-based backbones [25]. Following standard practice, distance $D(x, \tilde{x}; f_\theta) = 1 - \cos\big(f_\theta(x), f_\theta(\tilde{x})\big)$ is taken as the cosine distance between the CLS tokens taken from the last layer for DINO and MAE (before and after the layer normalization, respectively), and the embedding vector for CLIP and OpenCLIP. For LPIPS and DISTS, $D(x, \tilde{x}; f_\theta)$ is simply the distance metric itself. Given a triplet $(x, \tilde{x}_0, \tilde{x}_1)$, and a feature extractor $f_\theta$, the model vote is calculated as:

$$\hat{y} = \begin{cases} 1, & d_1 < d_0 \\ 0, & d_0 < d_1 \end{cases}, \text{where } d_0 = D(x, \tilde{x}_0; f_\theta) \text{ and } d_1 = D(x, \tilde{x}_1; f_\theta). \tag{1}$$

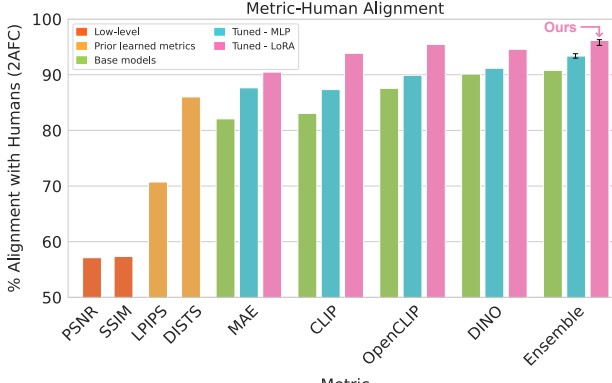

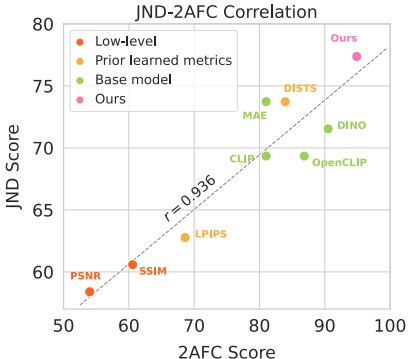

Figure 4: **Metrics performance on our benchmark.** Large vision models OpenCLIP and DINO outperform prior learned metrics LPIPS and DISTS (orange) (chance = 50%). Further tuning on our perceptual data with an MLP (blue) improves performance over out-of-the-box features (green); we find Low-rank LoRA [38] tuning boosts performance significantly (pink). Ensembling CLIP, OpenCLIP, and DINO models together improves performance as well. Our final model, *DreamSim*, combines these insights and achieves high agreement with humans (96.16%). Error bars represent a 95% confidence interval.

Figure 5: **Correlation between 2AFC & JND.** We plot the agreement between human judgments and existing similarity metrics on two tasks, 2AFC and JND. The strong correlation between the metrics' agreement with both tasks suggests that our dataset captures a general notion of similarity that can be replicated in two separate, independent human studies.

**Evaluating human-metric agreement.** Recall that for a triplet, we collect $y$, indicating which image a human selects as more similar. We evaluate how often each metric agrees with human judges as $\texttt{Score}_{\texttt{2AFC}}(f_\theta) = \mathbb{E}_{\mathcal{D}^{\text{2afc}}}[\mathbb{1}_{\hat{y}=y}]$ and $\texttt{Score}_{\texttt{JND}} = \mathbb{E}_{\mathcal{D}^{\text{jnd}}}[\mathbb{1}_{\hat{y}=s}]$.

## 4.2 Learning an improved metric

**Objective.** To better align a perceptual metric with human judgments, given triplet $(x, \tilde{x}_0, \tilde{x}_1)$ we maximize the difference between the perceptual distances $d_0, d_1$, with smaller distance associated with the distortion $y$ humans voted to be most similar. To accomplish this, we map $y \in \{0, 1\} \to \bar{y} \in \{-1, 1\}$ and use a hinge loss, equivalent to triplet loss [14] between the embeddings, with a margin of $m = 0.05$:

$$\mathcal{L}(y, \hat{y}) = \max(0, m - \Delta d \cdot \bar{y}), \text{where } \Delta d = \ d_0 - d_1. \tag{2}$$

**Tuning.** Next, we investigate the best method to tune or modify feature extractor $f_\theta$. A challenge is the large number of parameters (billions, for ViT backbones), compared to the relatively small amount of perceptual data. Inspired by [94], we first try tuning via an MLP head, adding a 1-hidden-layer MLP with a residual connection on top of the pre-trained embeddings. We compare this to fine-tuning through the pre-trained backbones using Low-Rank Adaptation (LoRA) [38], which we find to achieve better results than full fine-tuning. Using LoRA ($r = 16$, dropout $p = 0.3$, $\alpha = 0.5$) we tune approximately 0.67% of each model's parameters. In all experiments, LoRA significantly outperformed using the MLP head.

**Feature concatenation.** As multiple models may provide complementary features, we try combining them to boost performance. We concatenate features from the best-performing models on our dataset (DINO [10], CLIP [66], and OpenCLIP [17]) into one ensemble metric. By training just a handful of parameters in this ensemble (through LoRA [38]), we gain the benefits of each large model's embedding space without sacrificing much computational load while fine-tuning.

## 5 Experiments

### 5.1 How well do existing metrics align with human judgments?

In Figure 4, we show how well metrics align with humans on our dataset, across low-level metrics (PSNR, SSIM), prior learned metrics (LPIPS, DISTS) and recent large vision model embeddings

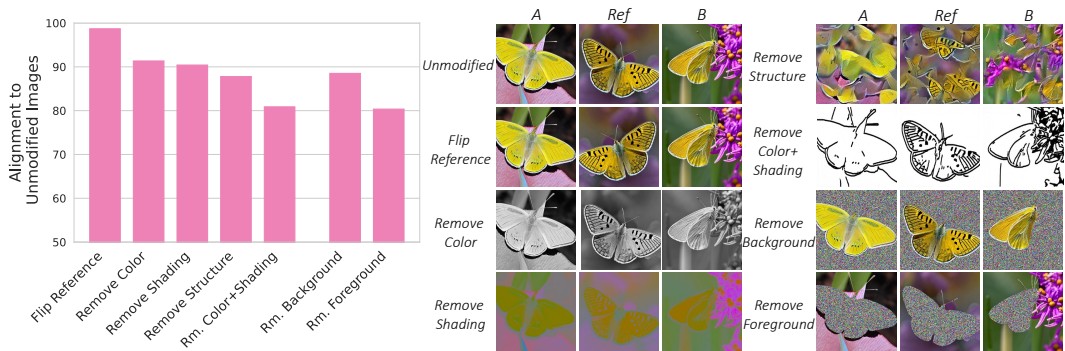

Figure 6: **Sensitivity of our similarity metric to different image ablations.** (Left) We ablate elements of the image, assessing performance dropoff by changing the orientation, color, shading, structure, and foreground vs. background content. The model is fairly robust to orientation, and more impacted by changes in color, shading, or structure. Foreground content is more important for similarity recognition than background content. (Right) For an example triplet (unmodified), we show visualizations of removing elements of the images.

(MAE, CLIP, DINO). The best-performing configurations of existing base models are DINO B/16, MAE B/16, CLIP B/32, and OpenCLIP B/32 (additional settings are in SM Section B.1). However, existing metrics have significant misalignments with humans; for example, DINO has approximately 10% disagreement. To improve performance, we finetune with LoRA (as described in Section 4.2), denoted as `Tuned - LoRA`, achieving significant alignment improvements over pre-trained baselines (Figure 4) and training an MLP head (`Tuned - MLP`). Ensembling models by concatenating Open-CLIP, CLIP, and DINO features achieves the highest score across baselines (90.8%), MLP-tuned models (93.4%), and LoRA-tuned models (96.2%). We refer to the best LoRA-tuned ensemble model as *DreamSim*, and use it for subsequent analysis and applications.

**Do different ways of measuring human perception agree?** We corroborate our results on our 2AFC test set by evaluating the models against JND perceptual judgments. The high correlation between 2AFC and JND scores indicates that our 2AFC judgments capture a general sense of similarity that can replicated across different settings (Figure 5).

**Does improved mid-level perception lead to improved low-level and high-level perception?** We use the same models as in Figure 4 to examine how performance on our dataset corresponds to performance on two others: BAPPS, containing image triplets with low-level augmentations, and THINGS, containing triplets with categorical variations. Training on our dataset improves human alignment on BAPPS but *decreases* alignment on THINGS, suggesting that humans, when making an automatic judgment, tend to focus on appearance-based visual differences (captured in BAPPS and our dataset) rather than high-level object category. We additionally evaluate on the TID2013 [64] and KADID-10k [49] image quality assessment (IQA) benchmarks as alternative low-level datasets. Similar to BAPPS, we find that our metric outperforms base models and is competitive with low-level metrics despite not training for low-level similarity. For full results, see SM Section B.1.

## 5.2 What image attributes affect similarity decisions?

**Sensitivity to image ablations.** We study which image properties affect our model's decisions by ablating properties of the image triplets, such as orientation, color, shading, structure, and foreground or background components. To do this, we modify the triplets to change or discard each attribute and compute the agreement between the model decisions on the modified and unmodified triplets. This operation measures the model's robustness when a given attribute is missing. Note that this analysis aims to provide insight into how our model determines similarity between two images, rather than to compare to human judgments; it is possible that humans may be more sensitive to different image attributes.

As shown in Figure 6, our model is least affected by orientation (tested by flipping the reference image), but ablating color or shading (the L or AB channels in LAB color space), structure [28], or both color and shading together using contours [12] has larger impacts. We conclude that color, shading, and structure contain critical information for our model, but our model can tolerate

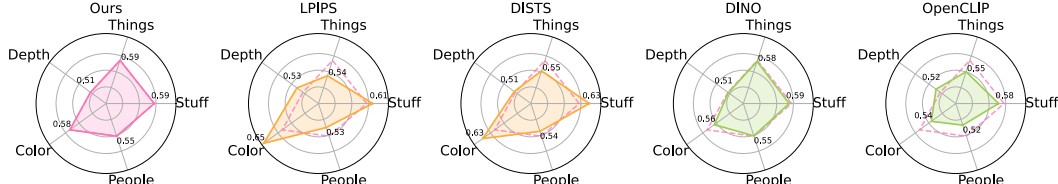

Figure 7: **Alignment with semantic and low-level metrics.** We select random triplets from the COCO dataset and assess how well low-level and semantic metrics predict model judgments. Compared to LPIPS & DISTS, our model is more sensitive to objects that appear in individual instances ("things") and the presence of people, and less sensitive to "stuff" categories, such as sky and road. Compared to OpenCLIP & DINO, color explains more of our model's decisions. We mark our model's results with a dashed pink line for comparison.

differences in orientation. To ablate foreground and background components, we segment the image [1] and replace the corresponding pixels with random uniform noise. This operation removes the color, shading, and texture of the region, even though the same outline remains in both. Removing the background has a smaller impact on model alignment compared to removing the foreground, suggesting that the foreground color and texture is more important for similarity recognition in our model compared to the background.

**Alignment with semantic and low-level metrics.** Given that some image attributes, like semantics, are difficult to ablate, we complement our ablation analysis by examining how well semantic and handcrafted low-level features align with model judgments. We use 10,000 random image triplets from MS-COCO [50] annotated with individual object instances ("things") and background categories such as grass and sky ("stuff"). We compare our metric to LPIPS, DISTS, DINO, and OpenCLIP (Figure 7). Our model aligns less strongly with RGB-color histogram similarity (calculated using histogram intersection [80]) compared to LPIPS and DISTS, however is more sensitive to color compared to other large vision models, while none align with depth map similarity [27, 69]. Next, we compute alignment with the things-category and stuff-category histograms, which summarize the area occupied by each semantic category. Our model aligns better with *things*-histogram similarity, whereas other models are comparatively more sensitive to *stuff*. This result suggests that our model is relatively more sensitive to foreground object instances. Finally, we compute how often metrics align with the *per-category* intersection of categorical histograms. Our model aligns best with the presence of people (additional results in SM Section B.1).

# 6 Applications

**Image retrieval.** We use our metric for image retrieval on two datasets: ImageNet-R [37], which contains 30K images of various renditions of ImageNet categories (e.g., sketches, cartoons), and a random subset of 30K images from MS-COCO [50], which contains diverse natural photos. Given a query image, we compute its similarity to the entire dataset and present the top-3 nearest images under each metric in Figure 8. As can be seen, our metric finds meaningful neighbors, demonstrating that it can generalize to diverse image domains. We compare to the best performing ViT embedding methods (OpenCLIP, DINO) and prior learned metrics (LPIPS, DISTS), as evaluated in Section 5.1. Compared to the baselines, which are either dominated by low-level similarity (e.g., background color in LPIPS/DISTS), or higher-level similarity (e.g., semantic class in OpenCLIP), our retrieved images exhibit similarity across a continuum of visual traits. See Section B.1 for more results and analysis on sketch-to-photo and photo-to-sketch domains.

We confirm our retrieval quality with a user study. For each metric, we retrieve the top-10 neighbors for a set of query images. For ImageNet-R, users prefer our metric's retrievals 36.8% of the time, followed by OpenCLIP (28.7%) and DINO (19.5%). We observe similar trends in a study with COCO images (see Figure 9).

**Feature inversion.** To better understand the information captured by our metric, we apply existing feature inversion techniques, where the aim is to optimize for an image that is similar to a target image under some metric. We compare our metric to the DINO and OpenCLIP embeddings, as well as to an ensemble embedding that simply concatenates all features without finetuning.

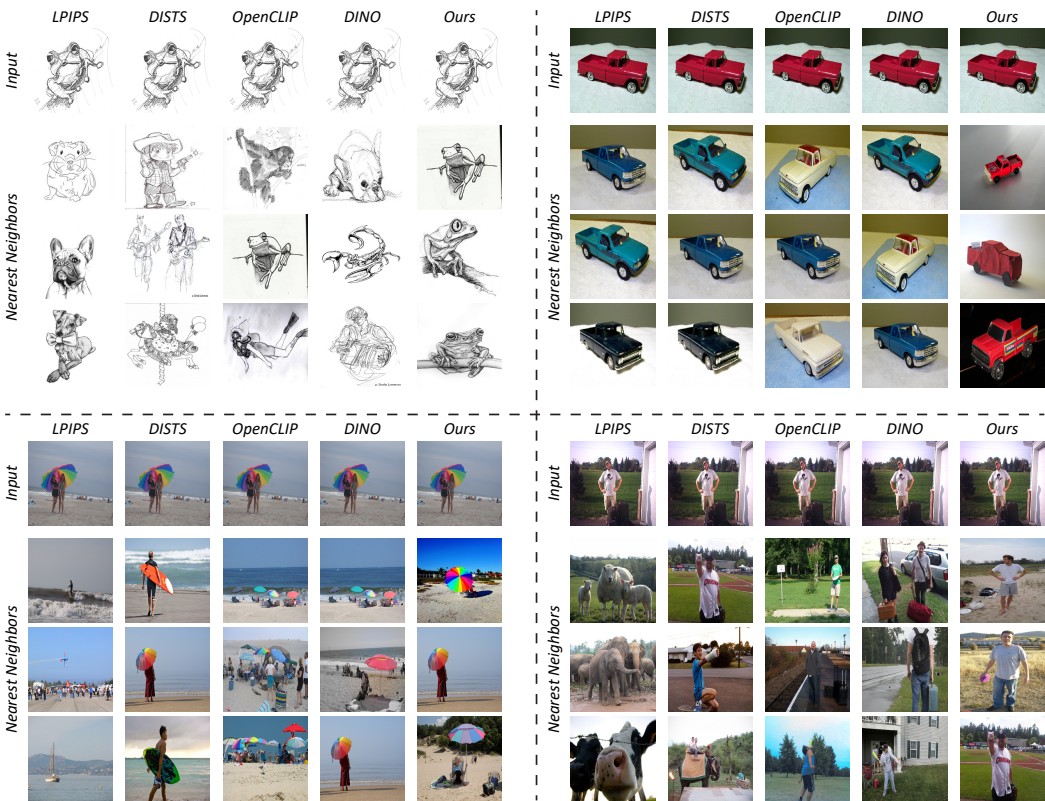

Figure 8: **Nearest-neighbor retrieval across different metrics.** Nearest neighbor retrieval on ImageNet-R (top) and COCO (bottom), comparing different metrics. Although the datasets include images outside of our training domain, our model consistently retrieves neighbors with similar appearance and class to that of the query image.

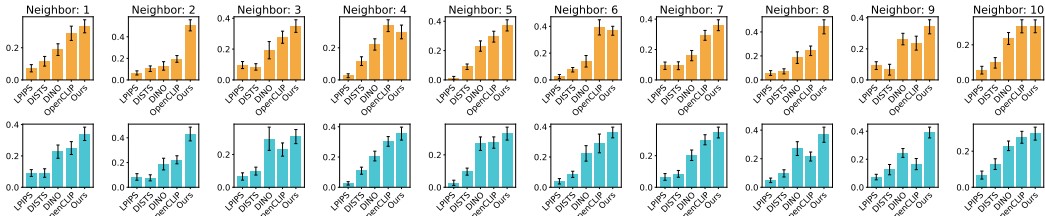

Figure 9: **User preferences for image retrieval results by metric.** We conduct a user study that collects preferences for retrieval results from LPIPS, DISTS, DINO, OpenCLIP, and DreamSim. We visualize one standard deviation above and below each bar. On ImageNet-R (top, orange), our metric is preferred by users 36.8% of the time, followed by OpenCLIP (28.7%). Similarly, on COCO (bottom, blue), users prefer our metric 33.6% of the time, with DINO being the second choice (28.1%).

Figure 10 shows the inversion results under a range of image priors. Under vanilla optimization, i.e., not using any image prior [52, 61], our metric's results exhibit higher fidelity to the original colors, shapes, and semantics. Using Deep Image Prior [83, 84], our metric reveals better global layout preservation and semantics. This is also evident when using our metric to guide recent generative models [22], resulting in higher quality image samples that resemble the semantic concepts and appearance of the target image.

Consistent with Section 5.2 and the retrieval results, these visualizations demonstrate our metric's sensitivity to color, layout, and semantics. The increased quality of our fine-tuned embeddings compared to just ensembling shows the effectiveness of tuning on our human-aligned dataset. See SM for more examples, and Section A.4 for details on feature inversion methodology.

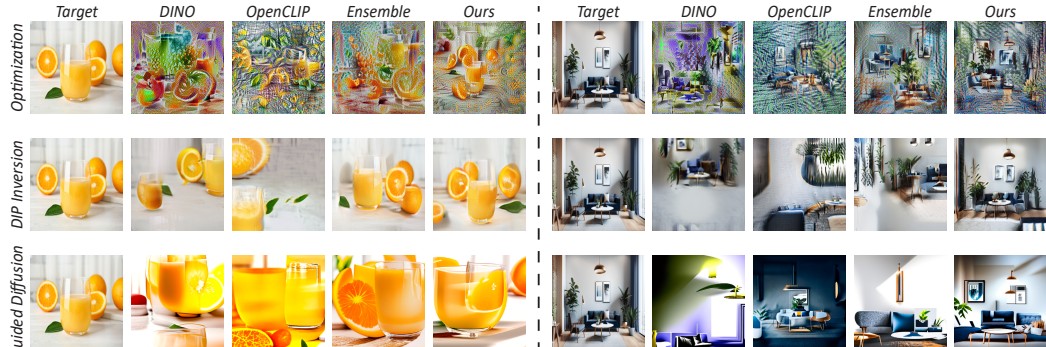

Figure 10: **Feature inversion across different metrics and image priors**. Given a target image, we optimize for an image, where the objective is to match the target image embedding of a given backbone. Without any image prior (Optimization), our metric better recovers the color, shape and semantics of the target image. With a weak image prior [83, 84] (DIP Inversion), our metric is able to reproduce scene structure and semantics. Using a diffusion model [22] as a strong prior, our metric better captures overall semantics and scene appearance.

**k-NN Classification.** We evaluate our metric as a *k*-Nearest Neighbors classifier, which requires the retrieval of images that are both visually and semantically relevant to a given query. We find that our metric outperforms all baselines on the ObjectNet [7] dataset and performs competitively with DINO on the ImageNet100 [81] dataset. For full details and results, refer to SM Section B.1.

# 7 Discussion

We expand the fundamental task of measuring perceptual image similarity to encompass factors beyond mere low-level similarity, spanning multiple notions of similarities (e.g., object poses, colors, shapes, and camera angles). By harnessing a state-of-the-art generative model, we overcome the lack of suitable data and introduce a new dataset consisting of human-judged synthetic image triplets. We evaluate candidate embeddings and propose a new similarity metric that better aligns with human judgments, demonstrating its generalization and effectiveness compared to existing metrics. We note that by using Stable Diffusion (SD) [71], our benchmark is exposed to potential preexisting biases and sensitive content in the model. Our perceptual model is also finetuned from existing pre-trained backbones, and thus may also inherit prior errors and biases. However, our work is a step in aligning representations to human preferences, and we believe our work can inspire further research in leveraging advances in generative image models for perceptual studies.

**Acknowledgments.** We thank Jim DiCarlo, Liad Mudrik, Nitzan Censor, Michelle Li, and Narek Tumanyan for fruitful discussions throughout the project. This work was supported by a Packard Fellowship to P.I., Israeli Science Foundation (grant 2303/20) to T.D., Meta PhD Fellowship to L.C., and NSF GRFP Fellowship to S.S.

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

# Appendix

In the supplemental materials, Sec. A contains additional methodological details on dataset collection and model training, Sec. B provides more analyses of our model, and Sec. C discusses the broader impact of our work, limitations, and licensing. Additional qualitative results, such as additional triplets from our dataset and visualizations of image retrieval and image reconstruction, are available on our project page.

## A  Method

### A.1  AMT Details

**User interface.** During the 2AFC study, users are instructed with the following prompt:

```
You will see three images:  one labeled "Reference",
   one labeled "Image A", and one labeled "Image B".
Select whether Image A or B is more similar to the Reference.
```

For each task, users see an image triplet with the reference in the center and distortions on either side (randomized). Each user completes 2 practice tasks, 50 real tasks, and 10 sentinels (randomly placed), averaging 3 minutes for the entire assignment. We discard responses from users who do not respond with 100% sentinel accuracy. See Figure 11 (left) for an example of the user interface.

Instructions for JND are as follows:

```
You will see four images one after the other.
Determine whether the first and third images are identical, then whether
        the second and fourth images are identical.
           Each correct answer earns 1 point.
```

Users are shown four images for 500 ms each with a 1 second gap inbetween, and are then prompted to answer the two questions in Figure 11 (right). They are given feedback and a score, though these have no bearing on the JND results themselves. Each user completes 2 practice tasks, 24 tasks with "different" pairs, and 12 tasks with "same" pairs, averaging 10 minutes for the entire assignment.

The object retrieval study instructs users with the following text:

```
You will see a reference image and five other images under it.
      Pick the image that is most similar to the reference.
```

See Figure 12 for an example of an object retrieval task. Each user completes 2 practice tasks, 40 real tasks, and 5 sentinels (randomly placed), averaging 3 minutes for the entire assignment. We discard responses from users who do not respond with 100% sentinel accuracy.

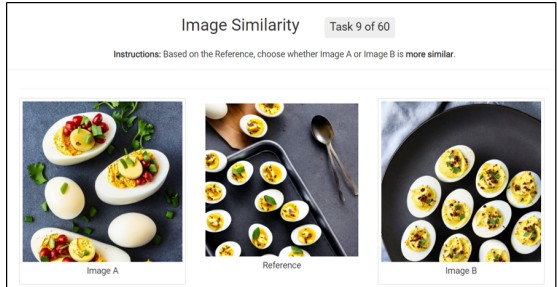 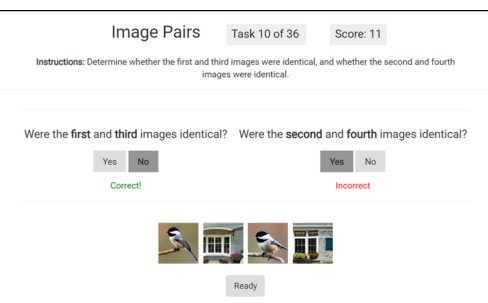

Figure 11: **User interface for AMT studies.**  (Left) One image triplet shown to a user in 2AFC, who is prompted to pick "Image A" or "Image B". (Right) In each JND task, users are shown a sequence of images and asked whether the image pairs were identical. Upon answering, they are given the correct answers.

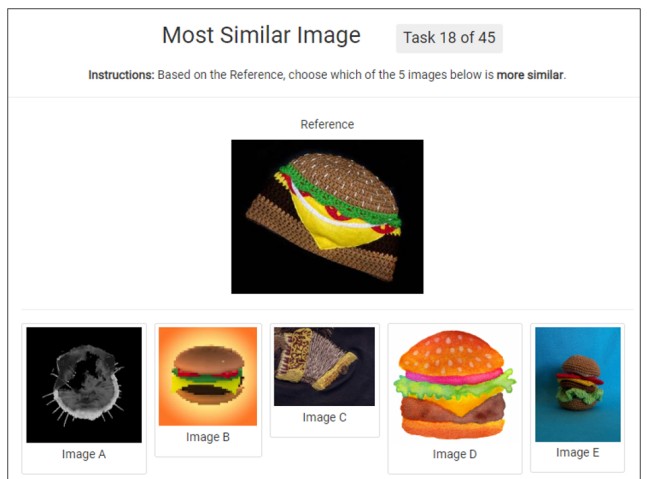

Figure 12: **User interface for image retrieval study.** To evaluate object retrieval performance, users are asked to pick the image (A-E) most similar to the reference.

| Round | # unanimous | # additional sentinel failures | # kept |
|---|---|---|---|
| 1 | 100,000 | 0 | 100,000 |
| 2 | 74,346 | 6,750 | 81,096 |
| 3 | 63,423 | 2,411 | 65,834 |
| 4 | 51,097 | 592 | 51,689 |
| 5 | 43,615 | 289 | 43,904 |
| 6 | 37,696 | 113 | 37,809 |
| 7 | 30,420 | 16 | 30,436 |
| 8 | 25,079 | 0 | 25,079 |
| 9 | 22,098 | 0 | 22,098 |
| 10 | 20,019 | 0 | 20,019 |

Table 2: **Filtering for cognitively impenetrable triplets.** We start with 100K triplets, and advance triplets to the subsequent round if the human vote remains unanimous, or if the added vote came from a user who did not pass the sentinels and thus the vote is inconclusive (the vote from this user is discarded).

**Cost breakdown.** Across 10 rounds of 2AFC studies, we show users 477,964 triplet instances (see Table 2 for the full breakdown). Each user is paid $0.50 for one assignment consisting of 50 triplets, averaging $10.00/hr. In total, we pay users $4779.64 to collect 2AFC judgments on our dataset.

We run JND on image triplets in our test set that received $\geq 6$ 2AFC judgments. Each user is paid $2.00 for one assignment consisting of 48 image pairs, averaging $12.00/hr. In total, we pay users $156.00 to collect JND judgments on our test set.

Our object retrieval user study is conducted over 200 query images with 10 nearest neighbors. Users are paid $0.50 for one assignment consisting of 40 tasks, averaging $10.00/hr. In total, we pay users $40.50 to collect object retrieval judgments.

## A.2    Additional Dataset Details

**Filtering text prompts.** As described in Sec. 3.1 we sample images using image labels drawn from popular image classification datasets. We perform basic filtering to avoid human categories that typically result in malformed generated images, such as "man" and "baby". Thus, human faces only feature in a small percentage of our dataset (<0.5%), typically in categories such as "lab coat" and "bonnet" that are still associated with humans. We note that even with some human categories

excluded, our model is still sensitive to the presence of people, as shown in Fig. 7 and Fig. 8. For a full list of categories used, refer to our GitHub page.

**2AFC task.** Following Sec. 3.2 in the main text, we filter the images over 10 rounds to obtain cognitively impenetrable triplets where humans tend to vote the same way despite various differences between the images. Statistics for each round of filtering is reported in Table 2, which leaves us with roughly 20% of the original triplets containing unanimous votes. We discard all votes from any worker who fails the sentinel task. As a result, not all triplets have the same number of votes. The resulting sizes of the train, validation, and test splits and additional statistics on each split are reported in Table 3 (top). In practice, we further discard triplets with five or fewer unanimous votes.

**JND task.** The JND triplets are meant to capture the decision boundary where two different images are similar enough to be confused as identical, in the presence of a masking image. Each triplet is divided into two pairs: Ref vs. A and Ref vs. B. These pairs are presented to different workers in different interleaving sequences. We collect three judgments for each pair and take the majority vote among the three judgments, thus collecting six judgments in total per triplet (Table 3 (bottom)).

| Dataset | Split | # Samples | Avg # Votes | Consensus Type |
|---------|-------|-----------|-------------|----------------|
| 2AFC | Train | 15941 | 7.11 | Unanimous |
|  | Validation | 1958 | 7.07 | Unanimous |
|  | Test | 2120 | 7.04 | Unanimous |
| JND | Test | 411 | 6 | Majority |

Table 3: **Dataset Statistics.** For each dataset, we report the number of samples and the average number of votes for each triplet after filtering for sentinel failures. Labels for the 2AFC dataset are based on a unanimous vote, while for JND we take the majority vote over three trials per pair (six trials per triplet).

## A.3    Model Training.

For all fine-tuned models (both `Tuned - MLP` and `Tuned - LoRA`) we use the NIGHTS training dataset, with an 80%-10%-10% split as described in Table 3, and images of size $768 \times 768$ resized to $224 \times 224$. We train on a single NVIDIA GeForce RTX 3090 or NVIDIA TITAN RTX GPU with an Adam optimizer, learning rate of $3e - 4$, weight decay of 0, and batch size of 512 (non-ensemble models) and 16 (ensemble models). In MLP-tuned training, we use a width of 512. We tune the number of training epochs using the validation set; for the `Tuned - LoRA` ensemble model (DreamSim) we train for 6 epochs. For `Tuned - LoRA` models we use rank $r = 16$, scaling $\alpha = 0.5$, and dropout $p = 0.3$. Training time is approximately 30 min/epoch for LoRA-tuned models, and 15 min/epoch for MLP-tuned models.

## A.4    Feature Inversion

In Fig. 10 we show examples of using our metric to optimize for an image generation that is similar to some target image. We do so in three experiments, with increasingly strong generative priors.

**Optimization.** In this setup, we initialize an image randomly and update it iteratively using gradient descent with respect to the distance between it and the target image. Given a random crop of our working image and the target image we compute the distance between the two using a specific metric (DINO/OpenCLIP/Ensemble/DreamSim) and their corresponding embeddings. Note that in this setup there is no generator neural network.

**Deep Image Prior.** This technique involves beginning with a frozen random noise which serves as an input to a trainable U-Net. The U-Net is optimized iteratively by comparing the output of the U-Net with the target image using different metrics and their embeddings. This introduces a CNN prior on the generated image as described in the Deep Image Prior paper [84].

**Guided Diffusion.** This technique involves using classifier guidance to guide the generation of a 256x256 unconditional ImageNet-trained diffusion model [23] using an input prompt image. The guidance is done by computing the gradient of some loss function, given the estimated clean image

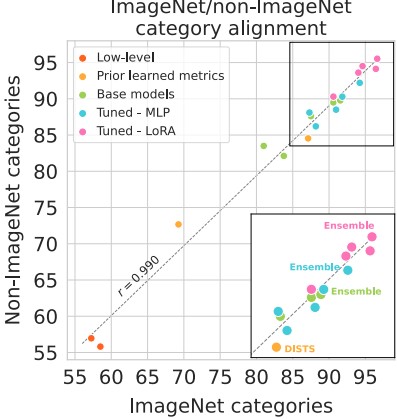

Figure 13: **Alignment on ImageNet and non-ImageNet triplets.** We split the test set into triplets generated from ImageNet categories and Non-ImageNet categories, as some model backbones are trained only on ImageNet images. For all models, alignment is highly correlated between the two splits.

and the target image, with respect to the current iteration. At every training step we compute the spherical distance loss between our model's embeddings of 64 random crops of the input image and the synthesized image. We combine this distance with a total variation loss, following [18, 58]. The gradient of this loss is then added to the mean of the current iteration, which pushes the generated image to be closer to the target image.

# B  Experiments

## B.1  Additional Evaluations

**Full Evaluation on Large Vision Models.** In Section 5.1 and Figure 4 in the main text we report results using the best-performing setting of various large vision model backbones. Table 4 shows the experimental variation over multiple runs, in which the LoRA variation consistently outperforms the MLP variation. Table 5 evaluates additional model settings, spanning different ViT model sizes, patch sizes, and strides.

| Ensemble tuning | Independent training seeds | | | | | Avg. | Stdev. |
|:---:|:---:|:---:|:---:|:---:|:---:|:---:|:---:|
| | Trial 1 | Trial 2 | Trial 3 | Trial 4 | Trial 5 | | |
| MLP | 93.4 | 93.1 | 93.1 | 93.9 | 92.9 | 93.3 | 0.326 |
| LoRA | 96.2 | 96.3 | 95.1 | 96.0 | 95.8 | 95.9 | 0.416 |

Table 4: **Experimental variation for ensemble models.** We train the `Tuned - MLP` and `Tuned - LoRA` ensemble models on 5 seeds each and record their test accuracies at the same epoch as was recorded in the main paper.

As some models are adapted from backbones trained on ImageNet [21] (including the prior learned metrics and DINO), we split our dataset into categories contained in ImageNet and those not in ImageNet, and evaluate alignment on each split. Performance on both splits is highly correlated (Figure 13), suggesting that the notions of visual similarity are related regardless of whether or not the triplet was generated from an ImageNet category, and whether or not the model was trained only on ImageNet.

We note that the MAE ViT B/16 checkpoint was taken from Hugging Face transformers repository [89], while the others were taken from the official Facebook repository [31].

**Alignment on Alternative Datasets.** As depicted in the left plot of Figure 14, training on our dataset (with either tuning method) indeed improves BAPPS metric-human alignment in nearly every model, suggesting that some of these patch-based distortions are implicitly still captured in our dataset. We

| Model | | | | Alignment | | |
|---|---|---|---|---|---|---|
| **Model Class** | **Model Name** | **Model Type** | **Feature** | **Overall** | **ImageNet** | **Non-ImageNet** |
| **Base Models** | PSNR | – | – | 57.1 | 57.3 | 57.0 |
| | SSIM | – | – | 57.3 | 58.5 | 55.8 |
| **Prior-Learned Metrics** | LPIPS | AlexNet-Linear | – | 70.7 | 69.3 | 72.7 |
| | DISTS | VGG16 | – | 86.0 | 87.1 | 84.5 |
| **Base Models** | CLIP | ViT B/16 | Embedding | 82.2 | 82.6 | 81.7 |
| | | ViT B/32 | Embedding | 83.1 | 83.8 | 82.1 |
| | | ViT L/14 | Embedding | 81.8 | 83.3 | 79.8 |
| | DINO | ViT S/8 | CLS | 89.0 | 89.7 | 88.0 |
| | | ViT S/16 | CLS | 89.6 | 90.2 | 88.8 |
| | | ViT B/8 | CLS | 88.6 | 88.6 | 88.5 |
| | | ViT B/16 | CLS | 90.1 | 90.6 | 89.5 |
| | MAE | ViT B/16 | CLS | 82.1 | 81.0 | 83.5 |
| | | ViT L/16 | CLS | 82.7 | 82.4 | 83.0 |
| | | ViT H/14 | CLS | 83.5 | 83.2 | 83.8 |
| | OpenCLIP | ViT B/16 | Embedding | 87.1 | 87.8 | 86.2 |
| | | ViT B/32 | Embedding | 87.6 | 87.5 | 87.6 |
| | | ViT L/14 | Embedding | 85.9 | 86.7 | 84.9 |
| | Ensemble | ViT B/16 | Mixed | 90.8 | 91.6 | 89.8 |
| **Tuned MLP** | CLIP | ViT B/32 | Embedding | 87.3 | 88.2 | 86.2 |
| | DINO | ViT B/16 | CLS | 91.2 | 91.8 | 90.3 |
| | MAE | ViT B/16 | CLS | 87.6 | 87.3 | 88.1 |
| | OpenCLIP | ViT B/32 | Embedding | 89.9 | 91.0 | 88.5 |
| | Ensemble | ViT B/16 | Mixed | 93.4 | 94.2 | 92.2 |
| **Tuned LoRA** | CLIP | ViT B/32 | Embedding | 93.9 | 94.0 | 93.6 |
| | DINO | ViT B/16 | CLS | 94.6 | 94.6 | 94.5 |
| | MAE | ViT B/16 | CLS | 90.5 | 90.6 | 90.3 |
| | OpenCLIP | ViT B/32 | Embedding | 95.5 | 96.5 | 94.1 |
| | Ensemble | ViT B/16 | Mixed | 96.2 | 96.6 | 95.5 |

Table 5: **Alignment on NIGHT test set.** We evaluate alignment on additional model settings, and separate the test set into ImageNet categories and non-ImageNet categories.

observe that MAE exhibits the best out-of-the-box performance, indicating a greater sensitivity to lower-level image distortions (e.g. color and shape) than DINO, CLIP or OpenCLIP. Surprisingly however, it is the only model whose performance decreases on BAPPS as it is further tuned. DINO, CLIP and OpenCLIP are not as sensitive to the image distortions in BAPPS, suggesting that before tuning, they are more attuned to higher-level image attributes that the dataset does not capture.

On THINGS, further training actually *diminishes* alignment with humans (see right plot of Figure 14). CLIP and OpenCLIP's superior performance on this dataset supports our hypothesis that they are more well-adjusted to higher-level image attributes, which THINGS aims to capture, rather than appearance-level variations.

Our evaluations across these three datasets show that, as we train perceptual metrics that align more closely with human perceptual similarity, we also improve on low-level similarity but perform slightly worse on high-level image distortions. These results suggests that humans, when making an automatic judgment, are more inclined to focus on immediate visual differences (captured in BAPPS and our dataset) rather than the image's category, context, or related words.

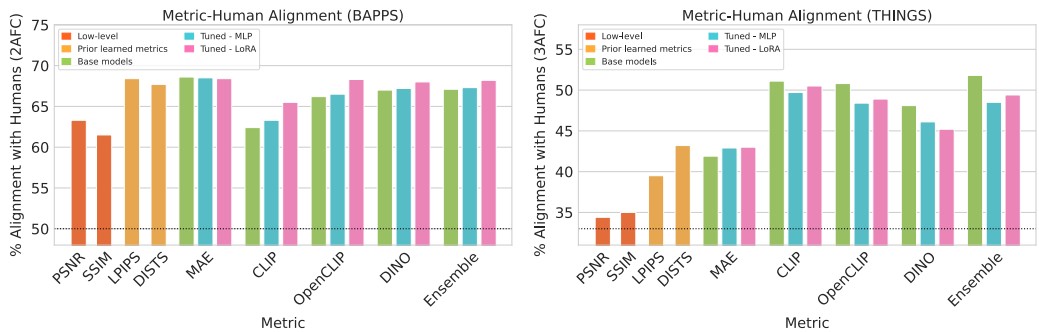

Figure 14: **Evaluation on existing low-level and high-level similarity datasets.** (Left) Despite never being trained for low-level similarity, LoRA-finetuned models on OpenCLIP, DINO, and Ensemble achieve similar human alignment to LPIPS, which was directly trained on the BAPPS dataset. (Right) The THINGS dataset measures high-level conceptual similarity, rather than appearance similarity. As such, we find that LoRA finetuning on our dataset degrades performance, as our triplets contain appearance similarity, by design.

| Metric | Image Part | Ours | DINO | OpenCLIP | DISTS | LPIPS |
|---|---|---|---|---|---|---|
| Color (RGB) | Foreground | 58.3 | 53.7 | 51.2 | 59.4 | 60.5 |
| Color (RGB) | Background | 58.7 | 57.2 | 55.5 | 64.1 | 64.7 |
| Color (RGB) | Total | 58.4 | 55.8 | 54.1 | 62.9 | 65.3 |
| Luminance | Foreground | 54.5 | 51.9 | 49.9 | 56.8 | 55.6 |
| Luminance | Background | 55.5 | 54.3 | 54.1 | 57.7 | 54.4 |
| Luminance | Total | 54.1 | 52.9 | 51.5 | 56.9 | 55.6 |
| Depth | Total | 54.2 | 53.6 | 53.3 | 54.7 | 55.8 |
| Category Histogram | Things | 58.7 | 58.3 | 55.1 | 55.3 | 53.8 |
| Category Histogram | Stuff | 59.5 | 58.8 | 57.9 | 62.5 | 61.3 |
| Presence of Person | - | 55.3 | 52.6 | 55.2 | 51.8 | 53.8 |
| Presence of Furniture | - | 53.1 | 52.2 | 54.2 | 53.4 | 53.6 |
| Presence of Textiles | - | 52.8 | 52.4 | 53.1 | 51.6 | 53.6 |

Table 6: **Automated Metrics on COCO.** Alignment of hand-crafted metrics with model decisions on the COCO dataset, which provides ground-truth semantic labels.

| Metric | Image Part | Ours | DINO | OpenCLIP | DISTS | LPIPS |
|---|---|---|---|---|---|---|
| Color (RGB) | Foreground | 71.7 | 70.0 | 69.3 | 68.7 | 60.6 |
| Color (RGB) | Background | 65.4 | 66.2 | 64.7 | 66.4 | 62.0 |
| Color (RGB) | Total | 69.8 | 67.9 | 67.6 | 66.0 | 64.3 |
| Luminance | Foreground | 63.1 | 62.6 | 62.2 | 61.4 | 56.1 |
| Luminance | Background | 59.3 | 59.5 | 58.8 | 60.3 | 56.8 |
| Luminance | Total | 59.4 | 60.6 | 59.8 | 59.2 | 56.5 |
| Depth | Total | 54.2 | 53.6 | 53.3 | 54.7 | 55.8 |

Table 7: **Automated Metrics on NIGHTS.** Alignment of hand-crafted metrics with model decisions on our dataset.

**Alignment with low-level features.** In Section 5.2 and Figure 7 of the main text we report results on the alignment between our metric, OpenCLIP, DINO, LPIPS, and DISTS with low-level and semantic metrics for the COCO dataset. In Table 6 we also report additional, fine-grained results for COCO triplets. We use CarveKit [1] to segment out the foreground and background of each image, and then break down how well each metric agrees with RGB color histogram similarity, luminance histogram similarity, and depth map distance for foreground, background, and the full image. For color histograms we use 32 bins for each channel, and for luminance histograms we use 10 bins.

We also examine semantic features. For each image, we find the percentage of area that each semantic category occupies, and then compute the alignment between the difference in area for each category and perceptual metrics. Note that when the difference in area is the same for both pairs in a triplet, it is counted as 50% alignment. When the difference in area is smaller for the pair chosen by the perceptual metric, it is counted as 100% alignment (and 0% in the case of disagreement). In Table 6 we show the five semantic categories most aligned with our metric. Our metric has a 55% alignment score with the "people" category, though does not align well above chance for other categories.

In Table 7 we show alignment with low-level metrics for our dataset (which does not have semantic annotations). On our dataset there is higher alignment with color, luminance, and depth across all metrics, as compared to COCO triplets. This is likely because the images in each of our dataset's triplets all share the same semantic category, making lower-level features more important than for the randomly-chosen COCO triplets. Our model aligns significantly better with foreground metrics – particularly foreground color – whereas LPIPS aligns slightly better with background.

**Sketch-photo image retrieval.** To further analyze perceptual metric performance on out-of-domain data, we evaluate sketch-to-photo and photo-to-sketch image retrieval on the Sketchy database [74]. First, we assign sketches as queries and photos as the search space. As seen in Figure 16 (left), DINO, DISTS, and LPIPS perform poorly on this task compared to OpenCLIP and DreamSim because they focus on color similarity, retrieving photos with predominantly white/gray backgrounds. This sometimes appears in DreamSim's results as well, hindering performance (Figure 15, top-left). Without this failure case, we return impressive results (Fig. 15, top-right) focusing on the subject appearance. We also evaluate retrieval in the other direction, using photos as queries and sketches as the search space. As shown in Figure 15 (bottom), DreamSim outperforms all other metrics in retrieving the main object while remaining sensitive to its size and location in the image.

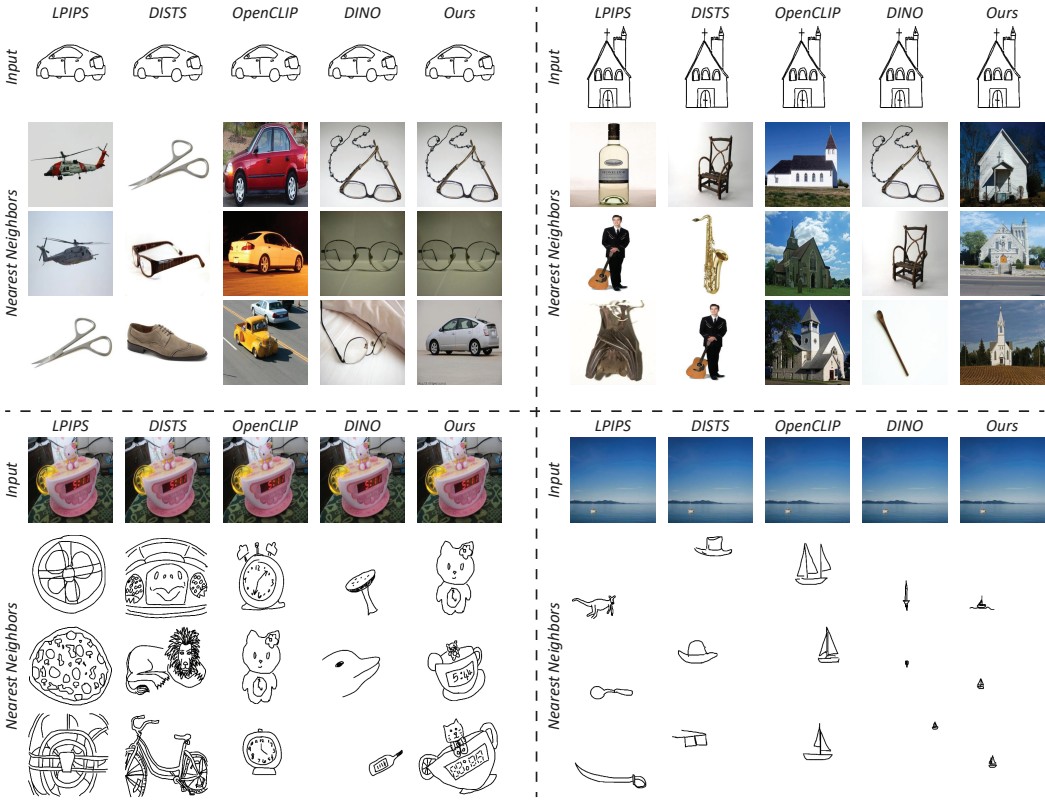

Figure 15: **Sketch-photo nearest-neighbor retrieval.** Image retrieval on the Sketchy database [74], with a sketch as query (top) and photo as query (bottom). OpenCLIP and DreamSim best preserve image content, returning photos of the subject itself (e.g. church, top right) rather than focusing on the black-and-white sketch style in the query. Our metric also best localizes the subject, returning a small boat in the corner while OpenCLIP and DINO both miss this spatial detail (bottom right).

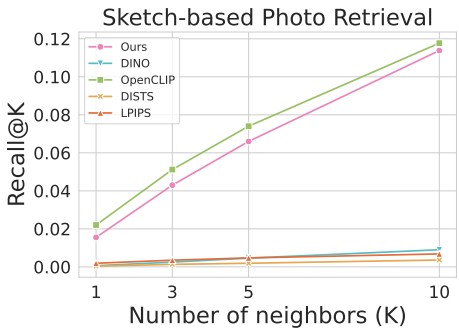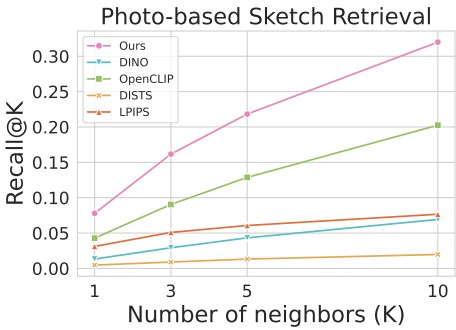

Figure 16: **Sketch-photo retrieval recall scores.** When using real photos as the queries, our model outperforms all methods across # nearest neighbors, obtaining a Recall@10 score of 0.32, compared to the next best model, OpenCLIP, with Recall@10 = 0.20. When using sketches as the queries, we find that our model is competitive with OpenCLIP and outperforms the remaining models.

**Image Quality Assessment (IQA) benchmarks.** We evaluate various metrics on the TID2013 [64] and KADID-10k [49] image quality assessment (IQA) benchmarks and report results in Table 8. Following standard IQA literature, we measure the Spearman Rank Order Correlation Coefficient (SROCC) between metric similarities and Mean Opinion Score (MOS). While LPIPS and FSIM remain best-performing overall, DreamSim is still competitive in IQA, outperforming most low-level metrics and base ViT models. Notably, our metric improves SROCC by $\sim 0.05$ after training on NIGHTS despite not having seen these low-level distortions. These results are consistent with those in Figure 14, where tuning on NIGHTS consistently improves human alignment on BAPPS.

|  |  | **TID2013** | **KADID-10k** |
|---|---|---|---|
| **Low-level** | PSNR* | 0.687 | 0.676 |
|  | SSIM | 0.720 | 0.724 |
|  | MS-SSIM | 0.798 | 0.802 |
|  | FSIM* | **0.851** | 0.854 |
| **Prior learned** | LPIPS | 0.838 | **0.883** |
|  | DISTS | 0.794 | 0.867 |
| **Base models** | DINO | 0.722 | 0.753 |
|  | OpenCLIP | 0.764 | 0.841 |
|  | Ensemble | 0.757 | 0.812 |
| **Ours** | DreamSim | 0.814 | 0.868 |

Table 8: **Evaluation on IQA datasets.** Alignment of various models on TID2013 [64] and KADID-10k [49]. While LPIPS and FSIM are best-performing overall, DreamSim is still competitive in IQA despite not having been tuned on low-level distortions. *While in some IQA literature PSNR/FSIM refer to computation on the luminance channel and PSNRc/FSIMc on color channels, we keep the metric names as-is and compute on color channels to be consistent with our PSNR labels elsewhere in the paper.

*k*-**Nearest Neighbors Classification.** We evaluate our model as a *k*-Nearest Neighbor (k-NN) classifier on the ObjectNet [7] and ImageNet100 [82] datasets, both standard classification benchmarks. Results are shown in Figure 17. Similarly to our other applications, we also evaluate OpenCLIP, DINO, and LPIPS as baselines (we were unable to evaluate DISTS due to computational constraints).

For ImageNet100 (a 100-category subset of ImageNet), we evaluate on the validation set and use the training set as the candidate search space. We note that DINO was trained using self-supervision on ImageNet, and has been previously shown to achieve strong k-NN results on ImageNet. DINO performs best, however our model still achieves competitive performance. For ObjectNet, we use an 80:20 split to randomly partition the dataset into a search-space and test set. Our model outperforms both OpenCLIP and DINO on top-1 classification across different values of $k \in \{1, 3, 5, 11, 21, 101, 201\}$.

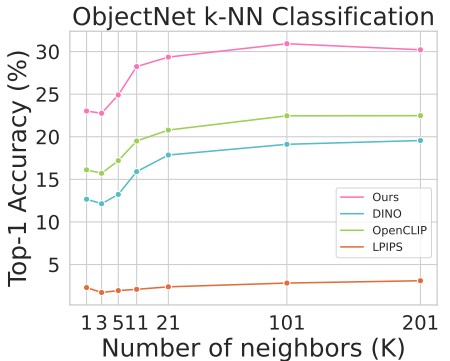
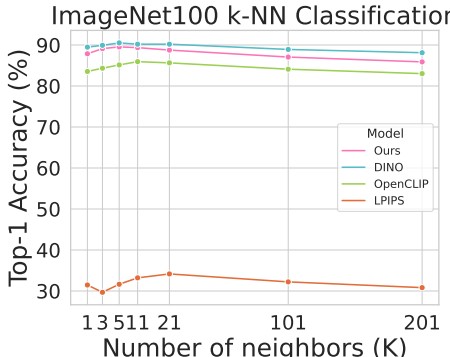

Figure 17: **k-NN classification results.** When evaluated as a k-NN classifier on the ObjectNet dataset, our model outperforms all methods across different values of $k$, with a maximum accuracy of 30.92%; the next best method, OpenCLIP, achieves 22.48%. When evaluating on ImageNet100, our model is competitive (1-2% difference) with DINO and outperforms other baselines.

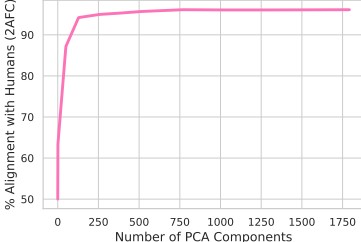

| # PCA Components | 2AFC Score |
|---|---|
| 1 | 63.4 |
| 512 | 95.7 |
| 768 | 96.1 |
| 1792 | 96.2 |

Figure 18: **Ablating Feature Dimension.** We apply a PCA decomposition to the output features of our model and vary the number of dimensions kept.

Table 9: **Feature PCA Decomposition.** We list 2AFC scores as a function of the number of PCA components kept, beating both the CLIP/OpenCLIP dimensionality (512) and DINO (768).

A common failure case for all three models is when the retrieved neighbors are visually similar to the query image but of a different category. We also note that ObjectNet is designed to differ from the ImageNet distribution, indicating that DreamSim may generalize well across different image distributions.

**Dimensionality reduction with PCA.** Our model consists of the concatenation of the DINO, Open-CLIP, and CLIP backbones, and therefore uses a higher-dimensional feature space to compute similarity compared to each of these models independently. To investigate whether the increased dimensionality is critical for improving human alignment, we ablate feature dimensions by applying PCA, taking a certain number of the top components, as seen in Figure 18 and Table 9. We can achieve comparable performance using just 500 of the top components, similar to the 512 dimensions of the CLIP and OpenCLIP embedding outputs, suggesting that the improved alignment is not just due to the higher-dimensional feature space used to compute similarity, but rather the additional capacity and model priors obtained from ensembling different models.

### B.2 Additional Visualizations

**Qualitative metric comparisons.** In Section 5.2 and Figure 6–7 of the main text we quantitatively analyze the differences between metrics. Here, we also provide a qualitative analysis by comparing image pairs that achieve high and low similarity scores for each metric.

In Figure 19 we plot, for each image pair in our dataset, the similarity scores from our metric against similarity scores from DINO, OpenCLIP, DISTS, and LPIPS. We show the same for image pairs drawn from the COCO dataset. In Figure 20 and Figure 21 we show the pairs where our metric most agrees and most disagrees with DINO, OpenCLIP, DISTS, and LPIPS, for our test set and the COCO dataset. Note that for COCO we draw random pairs of images that share at least instance category so

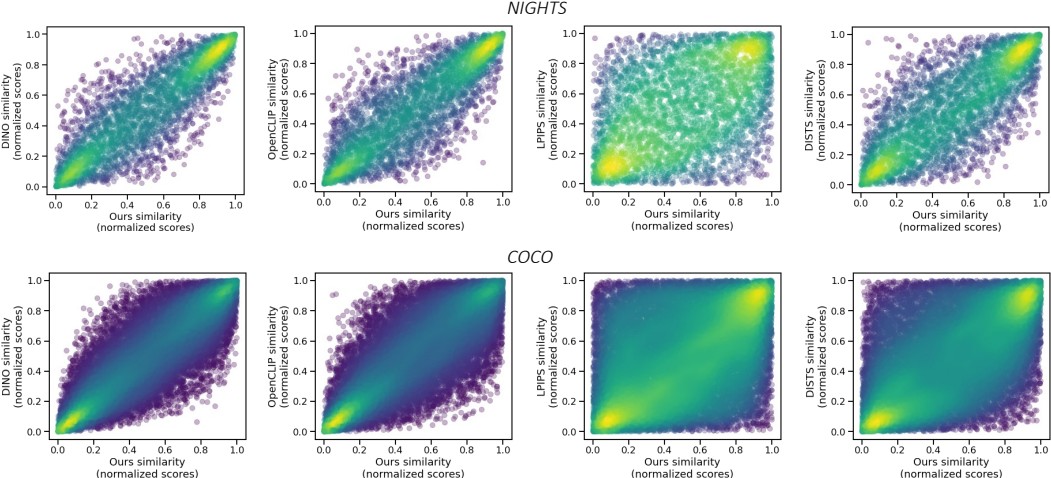

Figure 19: **Correlation between metric scores.** For image pairs from our dataset (above), and the COCO dataset (below), we plot similarity scores from our metric against similarity scores from DINO, OpenCLIP, LPIPS, and DISTS. Our metric's scores are most correlated with other ViT-based metrics, and correlate better with DISTS than LPIPS.

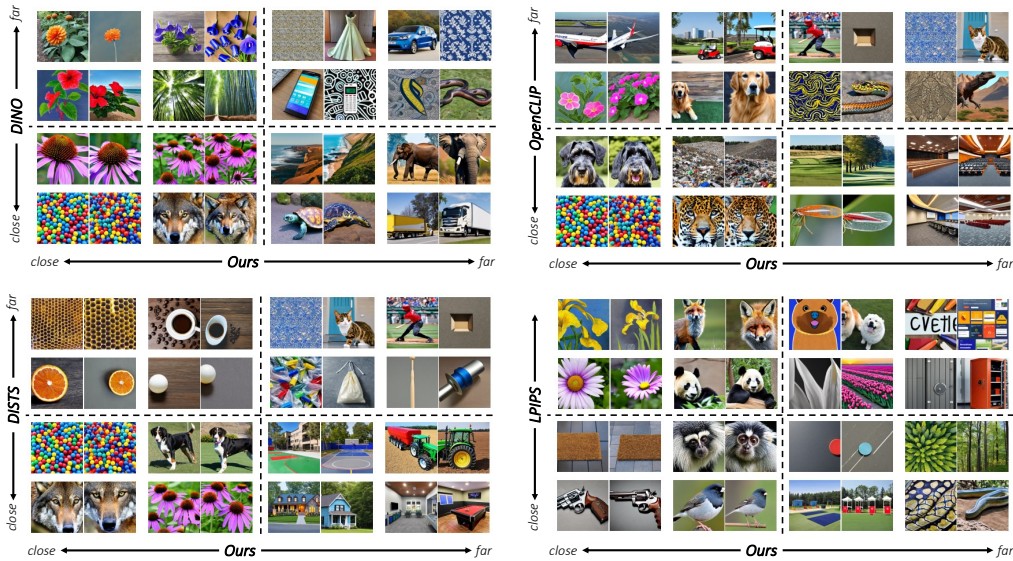

Figure 20: **Visualizing differences between our model and other metrics for NIGHTS.** We show the examples where our metric most agrees and disagrees with other metrics. Primary differences are that our metric is more sensitive to major changes in pose, semantics, and color. It is less sensitive to granular changes in structure when overall appearance is preserved, such as the honeycomb example in the DISTS quadrant and the forest example in the DINO quadrant.

that pairs have some semantic commonality, enabling better visualization of qualitative differences between metrics.

Comparing DISTS, the top-performing prior learned similarity metric, to our model, we find that DISTS is sensitive to structural changes despite similar overall appearance (such as the width of the honeycomb or the position of similar objects), while our model rates these pairs as nearby. On the other hand, pairs that are far in our feature space but close in DISTS feature space have less appearance similarity (*e.g.* the houses and rooms of different colors). Comparing to deep ViT features (DINO, OpenCLIP) our model is more likely to rate pairs with similar foreground color/appearance as similar, and less likely for pairs that are similar semantically but not appearance-wise. For COCO

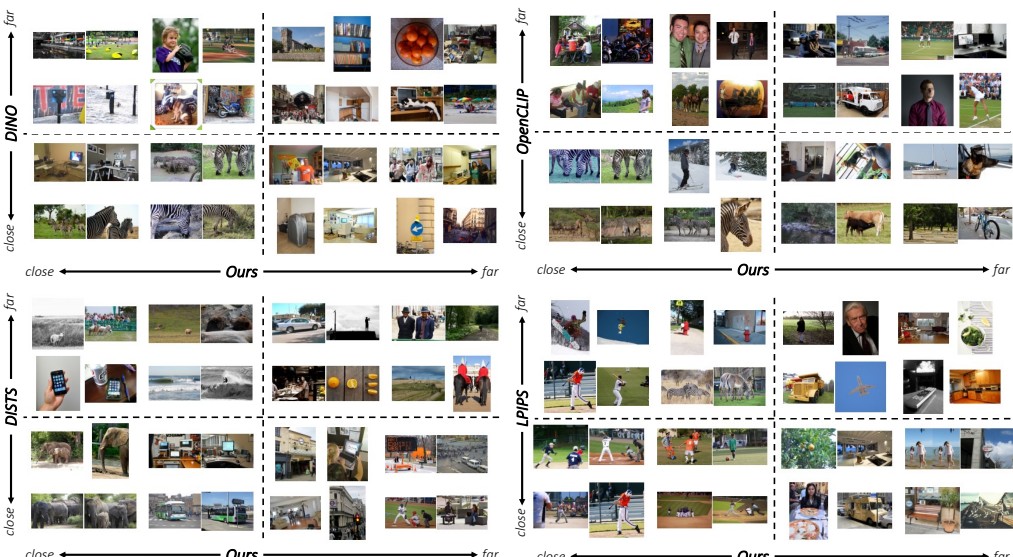

Figure 21: **Visualizing differences between our model and other metrics for COCO.** We show examples where our metric most agrees and disagrees with other metrics for pairs drawn from the COCO dataset. These pairs share fewer appearance similarities than pairs drawn from our dataset. Our metric seems particularly sensitive to foreground semantic similarities, such as the horse pair in in the OpenCLIP quadrant and the snowboarders in the LPIPS quadrant.

pairs, where there are fewer appearance similarities than in our dataset, our model chooses pairs that are similar semantically first, and only then appearance-wise.

**Attention Map Visualizations.** As an alternate way of understanding our model's similarity decisions, we visualize the transformer attention maps following Chefer et al. [15]. Our model consists of finetuned versions of DINO, CLIP, and OpenCLIP backbones, and the resulting attention map places the largest activations on the finetuned DINO backbone (Figure 22 (left)). Accordingly, the overall attention map looks largely similar to the attention map constructed from only the finetuned DINO branch. Compared to the pretrained versions of each model backbone, the finetuned model better captures full object identity (such as over the entire body of the fish), while also minimizing spurious attention values in the background (Figure 22 (right)). Consistent with earlier analysis, this supports the fact that the foreground plays a larger role in the DreamSim similarily judgment than the background.

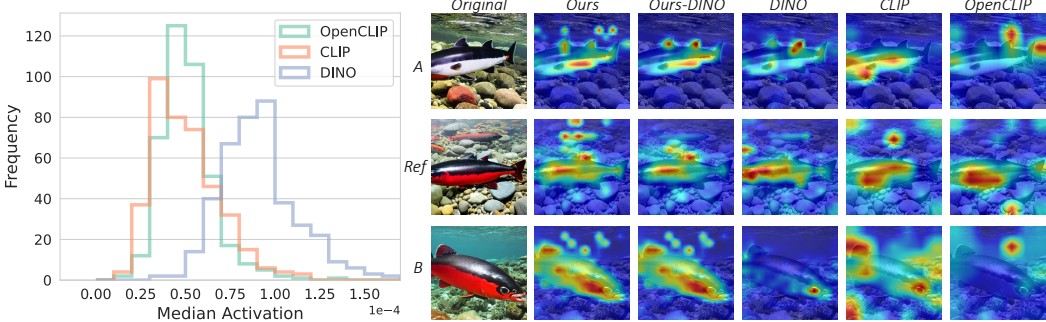

Figure 22: **Visualizing attention maps.** The finetuned DINO branch of our model has the largest contribution in the attention map [15], with the largest median activation compared to the CLIP and OpenCLIP branches (computed over 400 test set images). As such, in our model, the overall attention map is similar to the attention map from only the DINO branch. Compared to the pretrained backbones, our model better captures the entire object of interest (the fish body) while reducing spurious attention in background regions.

As the DINO model has the largest contribution in the attention maps, Figure 23 shows additional examples focusing on the difference between the finetuned DINO backbone within our model, and

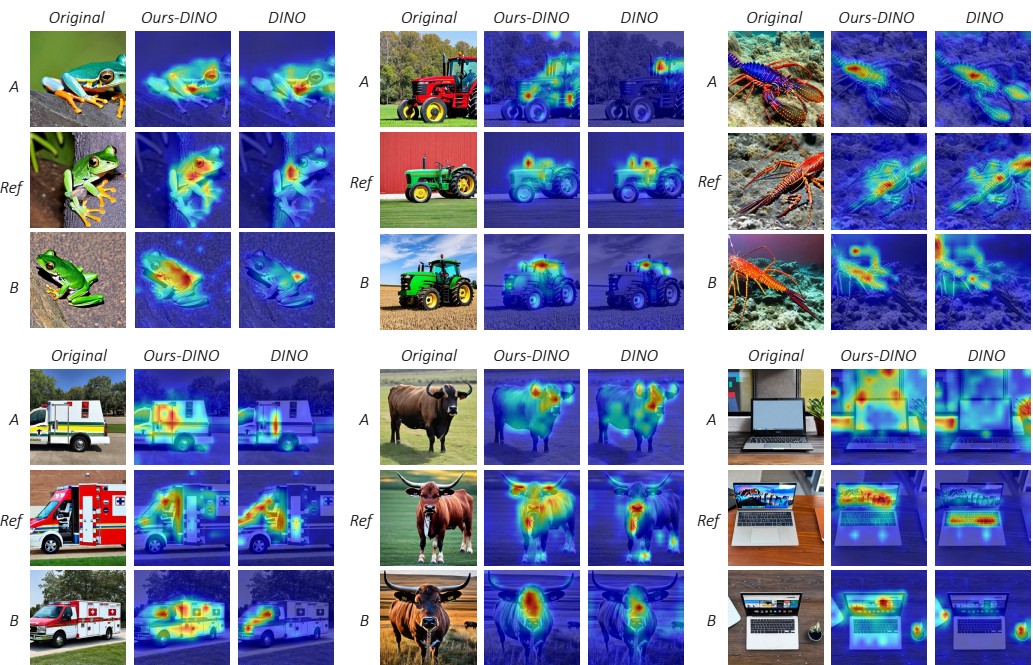

Figure 23: **Comparing our model and baseline attention maps from the DINO branch.** Focusing on the DINO branch, which has the largest contribution in our model's attention map, we compare the attention maps of the pretrained and finetuning models. The finetuned attention map in our model better captures the foreground object or relevant regions of interest. In all examples, the DINO baseline selects the A image, while humans and our model select the B image.

the pretrained DINO backbone prior to finetuning. This visualizes how the DINO backbone changes as it is finetuned on our dataset. Our finetuned model better captures the foreground object, while attention maps from the pretrained DINO backbone may only focus on small portions of the object. In the lobster example (Figure 23 (top-right)), our model places attention on the relevant parts of the object, such as the lobster body rather than the claws in the A image as the claws do not appear in the other two images.

## C Discussion

**Broader Impacts and Limitations.** Our model and dataset are developed from pretrained Stable Diffusion, CLIP, OpenCLIP and DINO backbones. As such, our model can inherit and propagate biases existing in these models for decisions on downstream tasks. Our dataset is generated using prompts that describe a single category, and we manually filter out categories that may generate sensitive content, such as violence or malformed human faces. As a result, the dataset has a large focus on object-centric domains, and content containing humans is considered out-of-domain. The resulting dataset and model does not capture the full range of human similarity judgements, but only the variations that we can capture in our synthetically-generated dataset.

**Licenses.** The icons for Figure 1 as well as the images for the inversion experiments are licensed by Adobe Stock, under the Adobe Stock Standard License, and by Pixabay, under their content license.

**IRB Disclosure.** We received IRB approvals for our AMT experiments from all of the institutions involved. Accordingly, we took measures to ensure participant anonymity and refrained from showing them potentially offensive content.

