# OpenReview forum: "DreamSim: Learning New Dimensions of Human Visual Similarity using Synthetic Data"
_NeurIPS.cc/2023/Conference — NeurIPS 2023 spotlight_

### Official Review · Reviewer_hsYj · 2023-06-13

**Soundness:** 4 excellent
**Presentation:** 4 excellent
**Contribution:** 4 excellent
**Rating:** 8
**Confidence:** 5

**Summary:**

This paper introduces a new dataset consisting of images generated by prompting the Stable Diffusion models, aiming for studying the mid-level image similarities. Human perceptual judgments are collected for image triplets via both 2AFC and JND tests. A perceptual metric is built upon large pretrained vision models, which is empirically shown to be highly consistent with human judgments.



**Strengths:**

+ It makes one of the first attempts to study the evaluation of AIGC images, the contribution of the new dataset is significant and the pipeline to collect human perceptual data is reasonable.
+ Experiments are well organized, which manifests the shortages of current perceptual metrics in evaluating AIGC images and verifies the effectiveness of the proposed metric.
+ The feature inversion part is interesting, aligning well with the Analysis-by-Synthesis model evaluation methodology.

**Weaknesses:**

- The definition of distortion seems a bit confusing. In the literature, distortions are generally associated with degradation of image quality. In this work, distortions refer to mid-level changes, which do not necessarily lead to the loss of visual quality.
- It is unclear how the JND experiment is conducted. When the participants are expected to give an answer of yes? Two images without any noticeable pixel changes, or perceptually the same in terms of the mid-level change?
- More details of the feature inversion part are expected.

**Questions:**

Does the proposed metric perform well on classical image quality assessment datasets?

**Limitations:**

Not applicable.

---

> ### Author Rebuttal · Authors · 2023-08-09
>
> We thank the reviewer for their comments. We are glad that the reviewer finds the contribution of our dataset to be significant, and the experiments well organized. Below we address specific questions and suggestions.
>
> **Definition of distortion**
>
> We use the term “distortions” to refer to variations between two images. We do so to be consistent with Zhang et. al [1], where distortions refer to visual changes relative to a reference image, particularly in the context of 2AFC experiments.
>
> In general we divide the spectrum of possible variations into low, mid, and high-level.
> - Low-level:, Pixel-level changes such as blurring or color shifts (As seen in BAPPS [1])
> - Mid-level: Variations in pose, spatial layout, style, numbers, etc. (As seen in NIGHTS)
> - High-level: Deals with conceptual information as opposed to perceptual in the low and mid levels, i.e knowledge that isn’t apparent in the pixels themselves. For example, a toaster and a skillet are conceptually similar as cooking appliances, though they don’t share much visual similarity (As seen in THINGS [2]).
>
> We will make these divisions more apparent in the final version of our paper.
>
> **JND Experiment**
>
> In the JND experiment, participants are shown 2 interleaved image pairs and asked whether the images in each pair were identical (i.e pixelwise) or not. We construct image pairs as $(x, \tilde{x})$ or $(x, x)$ so there is indeed a correct answer. Notably, this task is not simple due to the time between images, where participants see a blank screen for 1 second, and the interleaving of pairs designed to overcome the use of visual working memory.
>
> JND image pairs are created by splitting 2AFC image triplets $(x, \tilde{x}_0, \tilde{x}_1)$ into $(x, \tilde{x}_0)$ and $(x, \tilde{x}_1)$. Each pair receives either a “same” or “different” vote, and we take triplets receiving one of each to conclude that the $\tilde{x}$ associated with the “same” vote is more perceptually similar to $x$ than the $\tilde{x}$ associated with the “different” vote. We omit JND data from triplets that receive two “same” votes or two “different” votes as they provide no perceptual signal as to which distortion is more similar. We will clarify in the revision.
>
> **Feature Inversion**
>
> We perform the inversion experiment to better understand the information encoded in embeddings across different models. We do this with 3 separate experiments, with increasing prior strength.
>
> - Optimization - In this setup, we initialize an image randomly and update it iteratively using gradient descent with respect to the distance between it and the target image. Given a random crop of our working image and the target image we compute the distance between the two using a specific metric (DINO/CLIP/DreamSim) and their corresponding embeddings. Note that in this setup there is no generator neural network.
>
> - Deep Image Prior - This technique involves beginning with a frozen random noise which serves as an input to a trainable U-Net. The U-Net is optimized iteratively by comparing the output of the U-Net with the target image using different metrics and their embeddings. This introduces a CNN prior on the generated image as described in the Deep Image Prior paper [3].
>
> - Guided Diffusion  - This method involves using classifier guidance to guide the generation of an unconditioned diffusion model. The guidance is done by computing the gradient of some loss function, given the estimated clean image and the target image, with respect to the current iteration $x_t$. At every training step we take the spherical distance loss between our model's embeddings of 64 random crops of the input image and the synthesized image. We combine this distance with a total variation loss, following [4, 5]. The gradient of this loss is then added to the mean of the current iteration, which pushes the generated image to be closer to the target image.
>
> We will include these details in the revision.
>
> **Image Quality Assessment Benchmarks**
>
> We ran our model on TID2013 [6] and KADID-10k [7] datasets for image quality assessment. We discovered that DreamSim is competitive on this task, which aligns with our previous results on the BAPPS dataset. Additional details can be found in the global response.
>
> **References**
>
> [1] Zhang, Richard, et al. "The unreasonable effectiveness of deep features as a perceptual metric." Proceedings of the IEEE conference on computer vision and pattern recognition. 2018.
>
> [2] Hebart MN, Dickter AH, Kidder A, Kwok WY, Corriveau A, Van Wicklin C, et al. (2019) THINGS: A database of 1,854 object concepts and more than 26,000 naturalistic object images. PLoS ONE 14(10): e0223792. https://doi.org/10.1371/journal.pone.0223792
>
> [3] Ulyanov, Dmitry, Andrea Vedaldi, and Victor Lempitsky. "Deep image prior." Proceedings of the IEEE conference on computer vision and pattern recognition. 2018.
>
> [4] Mullis, Clay. “Clip Guided Diffusion.” https://github.com/afiaka87/clip-guided-diffusion
>
> [5] Crowson, Katherine. “K-Diffusion.” https://github.com/crowsonkb/k-diffusion
>
> [6] N. Ponomarenko, L. Jin, O. Ieremeiev, V. Lukin, K. Egiazarian, J. Astola, B. Vozel, K. Chehdi, M. Carli, F. Battisti, C.-C. Jay Kuo, Image database TID2013: Peculiarities, results and perspectives, Signal Processing: Image Communication, vol. 30, Jan. 2015, pp. 57-77.
>
> [7] H. Lin, V. Hosu and D. Saupe, "KADID-10k: A Large-scale Artificially Distorted IQA Database," 2019 Eleventh International Conference on Quality of Multimedia Experience (QoMEX), Berlin, Germany, 2019, pp. 1-3, doi: 10.1109/QoMEX.2019.8743252.

---

> > ### Comment · Reviewer_hsYj · 2023-08-11
> > **Post rebuttal**
> >
> > The authors have addressed my concerns well in the rebuttal, I am willing to raise my rating to 8.

---

### Official Review · Reviewer_71tx · 2023-07-04

**Soundness:** 3 good
**Presentation:** 3 good
**Contribution:** 3 good
**Rating:** 7
**Confidence:** 3

**Summary:**

In this paper, the authors develop a perceptual metric that assesses  images similarity holistically. They first collect a new dataset of human similarity  judgments over image pairs that are alike in diverse ways. During the dataset construction, the authors utilized Stable Diffusion to create synthetic data, aiming to model mid-level similarity. Based on the dataset, the authors observe that popular perceptual metrics fall short of explaining the new data, so they introduce a new metric, DreamSim, tuned to better align with human perception. Finally, authors applied this metric in image retrieval and showed that it can generalize to real images.

**Strengths:**

Human visual similarity is an important and practical topic in the domain of computer vision. It has many potential applications such as image retrieval. One of the major contributions of this paper is the new dataset (NIGHTS), which contains human similarity judgments of 20k synthetic image triplets. I believe this dataset will be useful for fellow researchers in the related domain. Another strength of this paper is the proposed similarity metric that captures how humans naturally perceive image similarity. I also appreciate the authors detailed analyses and comparison of the metric and its performance.

**Weaknesses:**

Though the paper is well written, and the topic is interesting and practical, the paper has several weaknesses. First of all, as the authors have mentioned, due to the potential biases and other limitations of Stable Diffusion, the authors deliberately avoided human faces during dataset construction. However, human faces are a common subject in general visual scenes and human perception of images containing human faces should be studied. Second, though the authors have applied the metric in image retrieval and used Figure 8 as a vivid proof to demonstrate that the proposed metric can be generalized to real images, I still have several concerns: first, is it possible to test the proposed metric on another benchmark dataset of real images, instead of the image retrieval task? Second, who are the users who rated the preference on the image retrieval results? More details of performance evaluation are needed. Third, not all metrics in Figure 4 are evaluated in the image retrieval task, why? Finally, while the authors avoided human faces in the proposed dataset, the models retrieved all images with human faces in the bottom right block in Fig 8. Does it show that the model focused on features other than human faces?

**Questions:**

1.	Why do the authors not compare all the metrics used in Figure 4 during the image retrieval task?
2.	Who are the users who rated the preference on the image retrieval results?
3.	Can the authors give a more comprehensive evaluation of the generalizability of the proposed dataset?


**Limitations:**

A more comprehensive evaluation of the generalizability of the dataset and metric is needed. Furthermore, humans are sensitive to faces from birth. Visual similarity on images with human faces is important.

---

> ### Author Rebuttal · Authors · 2023-08-09
>
> We thank the reviewer for their helpful comments, and address their questions and concerns below.
>
> **Inclusion of human faces**
>
> As the reviewer notes, humans are uniquely sensitive to faces. We believe that face similarity is important but merits its own specialized studies, and a general-purpose diffusion model is not the best face generator. There is a rich history of metrics that focus on face similarity/recognition [1-4], apart from existing literature on general perceptual similarity.
>
> We additionally clarify that we did not exclude all human faces. Although this was not the focus of our dataset, several of our categories lead Stable Diffusion to generate images with humans, such as “lab coat” and “bonnet”.  We will revise our supplementary material to clarify this. We specifically exclude categories such as man and baby which often result in malformed generated images. Thus, human faces only feature in a small percentage of our dataset (<0.5%).
>
> Finally, we note that even with some human categories excluded, our model is still sensitive to the presence of people. In Fig. 7 we evaluate DreamSim on image triplets drawn from the COCO dataset, and find that it is more sensitive to the presence of humans than other semantic categories. Additionally, in Fig. 8, as the reviewer mentions, our model retrieves images with humans wearing similarly-colored clothing.
>
> **Benchmarking on real images**
>
> In Fig. 4 of the SM we evaluate our model on BAPPS and THINGS, which are benchmark perceptual similarity datasets with real images. For analysis of these results please refer to section B.1.
>
> As a further benchmark of generalization to real images, we evaluate our model as a k-NN classifier on ObjectNet [5] and ImageNet100 [6]. Note that ObjectNet is designed to capture a different image distribution from ImageNet. On ObjectNet our model outperforms other methods, with a **Top-1 accuracy of 30.2% with k=201, compared to OpenCLIP (next best) with 22.48%**. On ImageNet DINO performs best (accuracy of 90.52% with k=5) however our model is still competitive with 89.54%. We provide additional details in the global response, and full quantitative results in the attached PDF.
>
> Finally, in response to reviewers Mq9Y and hsYj, we further evaluate our model on Image Quality Assessment datasets and show competitive performance against our baselines in the global response.
>
> We hope that our additional evaluations on benchmark datasets demonstrate the generalizability of our model to real images on diverse applications. We will include these additional results in our revision.
>
> **Image Retrieval Evaluation**
>
> Like our 2AFC and JND judgments, image retrieval preferences were collected on Amazon Mechanical Turk. Please refer to A.1. in the supplement for more image retrieval user study details, and B.3 for the results.
>
> **Models for Retrieval Evaluation**
>
> For our investigations of applications in retrieval and feature inversion we evaluate the best-performing models from Fig. 4. We exclude SSIM and PSNR because they were the worst-performing “low-level” (non-ViT) metrics, and we exclude CLIP and MAE because they were the lowest-performing transformer-based metrics.
>
>
> **References**
>
> [1] Deng, Jiankang, et al. "Arcface: Additive angular margin loss for deep face recognition." Proceedings of the IEEE/CVF conference on computer vision and pattern recognition. 2019.
>
> [2] Meng, Qiang, et al. "Magface: A universal representation for face recognition and quality assessment." Proceedings of the IEEE/CVF conference on computer vision and pattern recognition. 2021.
>
> [3] Parkhi, Omkar, Andrea Vedaldi, and Andrew Zisserman. "Deep face recognition." BMVC 2015-Proceedings of the British Machine Vision Conference 2015. British Machine Vision Association, 2015.
>
> [4] Schroff, Florian, Dmitry Kalenichenko, and James Philbin. "Facenet: A unified embedding for face recognition and clustering." Proceedings of the IEEE conference on computer vision and pattern recognition. 2015.
>
> [5] Andrei Barbu, David Mayo, Julian Alverio, William Luo, Christopher Wang, Dan Gutfreund, Josh Tenenbaum, and Boris Katz. Objectnet: A large-scale bias-controlled dataset for pushing the limits of object recognition models. In Advances in Neural Information Processing Systems 32, pages 9448–9458. 2019.
>
> [6] J. Deng, W. Dong, R. Socher, L.-J. Li, K. Li and L. Fei-Fei, ImageNet: A Large-Scale Hierarchical Image Database. IEEE Computer Vision and Pattern Recognition (CVPR), 2009.

---

> > ### Comment · Reviewer_71tx · 2023-08-11
> >
> > Thanks authors for the rebuttal. My questions are well addressed.

---

### Official Review · Reviewer_Mq9Y · 2023-07-06

**Soundness:** 4 excellent
**Presentation:** 4 excellent
**Contribution:** 3 good
**Rating:** 8
**Confidence:** 4

**Summary:**

The paper expands the fundamental task of measuring perceptual image similarity to encompass factors
 beyond low-level and high-level similarity. To achieve this, it leverages the recent advances in generative visual AI and generate synthetic images with text prompts. These synthetic images are expected to contain multi-faceted notions of similarity, which are then handed over to humans for A/B similarity test. The proposed model then learns from these human similarity ratings and is shown to uncover the emergent hidden factors defining visual similarity. In extensive evaluations, it is shown to surpass a number of baselines, including DINO and CLIP.



**Strengths:**

-- Research angle on visual similarity is timely, given its rather narrow scope of examinations in the past. Breakthrough on this angle can benefit vision community at large,.

-- Using synthetic images for visual similarity learning is a promising technical path forward given our increasing controllability on generating target visual contents. Instead of harnessing human ratings on the similarity between any two real images and letting models to learn as a black-box, using synthetic images has the advantage of leaving the discretion to human designers on what constitutes visual similarity -- for example in this paper, although only superficially, manipulates on mid-level visual variations.

-- Using LoRA to efficiently tune pre-trained visual ensembles towards a visual similarity judge is also reasonable.

-- Extensive analysis is extensive.

**Weaknesses:**

The paper is itself is quite intact by itself. So what I propose here is not necessarily a weakness, but rather with such a new tool on visual similarity, I keep wondering what can it, immediately, benefit vision tasks at my hand. The paper showcases a pilot study on the application of image retrieval, but in my understanding, the paper's contribution is best leveraged as a universal (differentiable) plug-n-play module to provide a complementary power to many computer vision tasks.

For example, it would be lovely to see how this paper opens new opportunities to image quality assessment community, where the intrinsic ambiguity of ground-truth human ratings is painstaking for generalisable modelling. Can DreamSim help alleviate this problem as a regularisation force by requiring similar visual contents to be of similar quality score? Same for the problem of sketch-based image retrieval community, where sketch-image ground-truth paired data is ill-defined. In the work of [A], they have shown how leveraging a visual similarity metric like LPIPS can greatly advance the understanding of the problem. Can DreamSim bring even further breakthrough?

[A] Photo Pre-Training, But for Sketch. CVPR 2023


**Questions:**

- Would be great if the paper shows more practical use cases that benefit from such a new visual similarity judge. For example, as a plug-n-play module in diverse image retrieval tasks? Or as an essential component for better evaluating a generative visual agent?

---

> ### Author Rebuttal · Authors · 2023-08-09
>
> We thank the reviewer for the helpful suggestions, and are glad that the reviewer finds the analysis extensive and our contribution beneficial to the vision community. We address specific suggestions below.
>
> **Image Quality Assessment**
>
> The reviewer mentions that our model may be useful for varied vision tasks, including as a differentiable module and for image retrieval. To assess its potential for the domains cited by the reviewer, we evaluated on standard benchmarks. We first ran our model on the TID2013 [1] and KADID-10k [2] datasets for image quality assessment. We discovered that DreamSim is competitive on this task, which aligns with our previous results on the BAPPS dataset. Additional details can be found in the global response.
>
> **Sketch-Image Retrieval**
>
> We also ran our model on the Sketchy Database [3], which contains images and a set of corresponding sketches for each image. When using real photos as the queries, we find that our model outperforms all methods, obtaining a **Recall@10 score of 0.32, compared to the next best model, OpenCLIP, with a score of 0.20**. When using sketches as the queries, we find that our model is competitive with OpenCLIP and outperforms the remaining models. We find that a common failure case for our model when using a sketch query is that it finds images of unrelated small objects on a white background to be more similar to the sketch, as our model relies on appearance similarity. Additional details can be found in the global response. Quantitative results and qualitative examples are provided in the attached PDF.
>
> **Practical Use Cases**
>
> Our sketch-image retrieval and IQA results demonstrate the potential for our model to be directly plugged into other retrieval pipelines, including images that are greatly different from the training domain. In our global response, we also investigate the applicability of our model to ObjectNet and ImageNet100 classification. We hope that these additional evaluations demonstrate that our model can be readily adopted into a diverse set of downstream tasks, and we will include these additional results in our revision.
>
> **References**
>
> [1] N. Ponomarenko, L. Jin, O. Ieremeiev, V. Lukin, K. Egiazarian, J. Astola, B. Vozel, K. Chehdi, M. Carli, F. Battisti, C.-C. Jay Kuo, Image database TID2013: Peculiarities, results and perspectives, Signal Processing: Image Communication, vol. 30, Jan. 2015, pp. 57-77.
>
> [2] H. Lin, V. Hosu and D. Saupe, "KADID-10k: A Large-scale Artificially Distorted IQA Database," 2019 Eleventh International Conference on Quality of Multimedia Experience (QoMEX), Berlin, Germany, 2019, pp. 1-3, doi: 10.1109/QoMEX.2019.8743252.
>
> [3] Sangkloy, Patsorn, et al. "The sketchy database: learning to retrieve badly drawn bunnies." ACM Transactions on Graphics (TOG) 35.4 (2016): 1-12.

---

### Official Review · Reviewer_23q7 · 2023-07-07

**Soundness:** 3 good
**Presentation:** 3 good
**Contribution:** 3 good
**Rating:** 7
**Confidence:** 3

**Summary:**

This paper leverages recent advances in synthetic image generation (namely text-to-image) to explore the space of image similarity assessment. They generate a novel dataset based on human forced choice input on pairs of synthetic images generated with respect to a reference image as well as interleaved visual memory tasks to assess when images are noticeably different on first assessment. The metric they learn over this novel dataset appears to better capture mid-level features such as pose and viewing angle.


**Strengths:**

The paper is overall clearly written. The experiments are well-described and compelling, with multiple complementary facets supporting each other. The paper provides both a dataset and metric which may be of interest to the image similarity domain.

**Weaknesses:**

I apologize for a somewhat unclear criticism, but the JND task appears to be a visual working memory task, not a perception task. As we know from investigations of change blindness, there can be quite distinct differences between images that humans will only notice given sufficient time to explore the image, but if they happen to be attending the location that is different, they will immediately notice the difference. This seems to be getting at a different aspect of similarity than the perceptual choice task, which makes me want more of an explanation for why these two tasks are paired, or how they interrelate and why they are paired in this way.

**Questions:**

- How are the categories drawn from the labels? Is it a random sampling (and if so, are duplicate categories controlled for?), a subset (and if so, how was this selected?), or another arrangement?

- Although it is mentioned that 7 votes is the average for retained triplets (L148), was there a minimum number required to be retained?

- How do humans react to the image ablations tested in Section 5.2? It seems unclear to me whether the model should be impervious to some of the changes made.

**Limitations:**

The limitations seem clearly articulated.

---

> ### Author Rebuttal · Authors · 2023-08-09
>
> We thank the reviewer for their helpful comments. We address questions and concerns from the reviewer below.
>
> **Reasoning for JND study**
>
> To clarify our purpose for JND, we pair it with 2AFC following prior works like Zhang et al. [1] and Nazeri et al. [2] to verify if our 2AFC annotations generalize to a separate, independent way of measuring similarity. A high correlation between 2AFC and JND scores indicates that our judgments are reliable under multiple experimental settings. As shown in Figure 5, we achieve $r^2=0.922$ across metrics, suggesting that despite the differences in the two tasks, there is a core notion of similarity shared between the tasks – two images that appear more similar when observed simultaneously are more likely to be confused with one another when observed briefly at different times.
>
> Additionally, JND is complementary as it is an objective measure of perceptual similarity; unlike 2AFC, which relies on subjective human ratings, it has a ground truth answer (images are either identical or different). However, JND is less data efficient than 2AFC as it relies on human mispredictions, whereas for 2AFC we can iteratively filter for a consensus agreement – thus it is easier to collect more data for 2AFC than JND.
>
> **How are categories drawn from labels?**
>
> We begin by collecting a list of all of the labels taken from the ImageNet, CIFAR-10, CIFAR-100, Oxford 102 Flower, Food-101, and SUN397 datasets. Then, we remove duplicates by transforming our list into a set. Finally, we remove labels that produce low-quality images (corresponding to obscure classes or complex objects).  This leaves us with a filtered set of 1449 labels, with ~55% contained in ImageNet. These labels are included in the file classes.txt in our supplementary material.  Our categories for our prompts are then simply randomly sampled from our filtered set to obtain 100,000 total triplets.
>
> **Minimum number of votes**
>
> Our NIGHTS dataset contains an average of 7 votes per triplet. For training, validation, and testing, we filtered for only those triplets with at least 6 votes to reduce noise.
>
> **Humans ratings on image ablations?**
>
> The ablation analysis in Section 5.2 does not claim that our model should be impervious to certain ablations; rather, we intend to understand what image attributes our model is sensitive to. This experiment gives us insight into how our trained model determines similarity between two images. Studying whether humans are sensitive to certain ablations and how models align to those judgments would be an interesting analysis of what models “should” be robust to, but we note that it is a separate experiment from our experiments in Section 5.2.
>
>
> **References**
>
> [1] Zhang, R., Isola, P., Efros, A.A., Shechtman, E., Wang, O.: The unreasonable effec- tiveness of deep features as a perceptual metric. In: IEEE Conference on Computer Vision and Pattern Recognition. pp. 586–595 (2018)
>
> [2] Nazeri, Kamyar, et al. "Edgeconnect: Generative image inpainting with adversarial edge learning." arXiv preprint arXiv:1901.00212 (2019).

---

### Author Rebuttal · Authors · 2023-08-09

We thank all reviewers for their feedback. We are glad they found our research well-motivated [Mq9y, 71tx].  Additionally, we appreciate their acknowledgment of our synthetic dataset and metric as novel [23q7, 71tx, hsYj], and of our experiments and analyses as well-organized and rigorous [23q7, Mq9Y, 71tx, hsYj].

We provide results on three additional applications based on suggestions from reviewers.

**Image Quality Assessment**

We ran our model on TID2013 [1] and KADID-10k [2] datasets for IQA. We note that these image distortions are low-level, structure-preserving changes, while our NIGHTS dataset does not preserve structure entirely but maintains appearance similarities. Following standard IQA literature, we measure the Spearman Rank Order Correlation Coefficient (SROCC) between metric similarities and Mean Opinion Score (MOS) [1,2,3] and obtain the results below, with best and second-best results bolded and italicized respectively:

|  |  | **TID2013** | **KADID-10k** |
|---:|---|:---:|---|
| **Low-level** | **PSNR*** | 0.687 | 0.676 |
|  | **SSIM** | 0.72 | 0.724 |
|  | **MS-SSIM** | 0.798 | 0.802 |
|  | **FSIM*** | **0.851** | 0.854 |
| **Prior learned** | **LPIPS** | _0.838_ | **0.883** |
|  | **DISTS** | 0.794 | 0.867 |
| **Base models** | **DINO** | 0.722 | 0.753 |
|  | **OpenCLIP** | 0.764 | 0.841 |
|  | **Ensemble** | 0.757 | 0.812 |
| **Ours** | **DreamSim** | 0.814 | _0.868_ |

While LPIPS and FSIM remain best-performing overall, DreamSim is still competitive in IQA. Additionally, our metric improves SROCC by ~0.05 after training on NIGHTS despite not having seen these low-level distortions. These results are consistent with those in Figure 4 of the supplement, where tuning on NIGHTS consistently improves human alignment on BAPPS.

*note that while in some IQA literature PSNR/FSIM refer to computation on the luminance channel and PSNRc/FSIMc on color channels, we keep the metric names as PSNR/FSIM and compute on color channels to be consistent with our PSNR labels elsewhere in the paper.

**KNN**

We evaluate our model as a k-NN classifier on the ObjectNet [4] and ImageNet100 [5] datasets, both standard classification benchmarks. k-NN classification requires retrieval of images that are both visually and semantically relevant to a query. Results are shown in Fig. 2 of the attached PDF. We also evaluate OpenCLIP and DINO as baselines. Due to time and computational constraints we were unable to complete evaluations for LPIPS and DISTS, however plan to include them in our revision.

For ImageNet100 (a 100-category subset of ImageNet), we evaluate on the validation set and use the training set as the candidate search space. We note that DINO was trained using self-supervision on ImageNet, and has been previously shown to achieve strong k-NN results on ImageNet. DINO performs best, however our model still achieves competitive performance. For ObjectNet, we use an 80:20 split to randomly partition the dataset into a search-space and test set. Our model outperforms both OpenCLIP and DINO on top-1 classification across different values of k.

A common failure case for all three models is when the retrieved neighbors are visually similar to the query image but of a different category. We also note that ObjectNet was designed to differ from the ImageNet distribution, indicating that DreamSim may generalize well across different image distributions.

**Sketch-Based & Photo-based Image Retrieval**

We ran two image retrieval experiments on the Sketchy database [6], which consists of photos and corresponding sketches.

In the first experiment, we assign sketches as queries and photos as the search space. As seen from Fig. 2a (attached), DINO, DISTS, and LPIPS perform poorly on this task compared to OpenCLIP and DreamSim (ours) because they focus on color similarity, retrieving photos with predominantly white/gray backgrounds. This sometimes appears in DreamSim’s results as well, hindering performance (Fig. 1 attached, top-left). Without this failure case, we can return impressive results (Fig. 1 attached, top-right). We thus hypothesize that our performance is impacted by generic photos in the search space that resemble sketches.

In our second experiment, we use photos as queries and sketches as our search space. Results are shown in Fig. 2b attached, showing that DreamSim outperforms all other metrics. Qualitatively, we handle complex photos well (Fig. 1 attached, bottom-left), and are sensitive to location and size of the foreground (Fig. 1 attached, bottom-right). This observation is consistent with the analyses in our main paper.

The results of these experiments together with those in the paper, show that DreamSim is competitive or performs better than all other benchmarks compared against, across a wide variety of tasks, ranging from low-level to high-level.

**References**

[1] Ponomarenko, Nikolay, et al. "Image database TID2013: Peculiarities, results and perspectives." Signal processing: Image communication 30 (2015): 57-77.

[2] Lin, Hanhe, Vlad Hosu, and Dietmar Saupe. "KADID-10k: A large-scale artificially distorted IQA database." 2019 Eleventh International Conference on Quality of Multimedia Experience (QoMEX). IEEE, 2019.

[3] Sheikh, Hamid R., Muhammad F. Sabir, and Alan C. Bovik. "A statistical evaluation of recent full reference image quality assessment algorithms." IEEE Transactions on image processing 15.11 (2006).

[4] Barbu, Andrei, et al. "Objectnet: A large-scale bias-controlled dataset for pushing the limits of object recognition models." Advances in neural information processing systems 32 (2019).

[5] Caron, Mathilde, et al. "Emerging properties in self-supervised vision transformers." Proceedings of the IEEE/CVF international conference on computer vision. 2021.

[6] Sangkloy, Patsorn, et al. "The sketchy database: learning to retrieve badly drawn bunnies." ACM Transactions on Graphics (2016).

---

### Decision · Program_Chairs · 2023-09-21

**Decision:**

Accept (spotlight)

**Comment:**

The paper proposed a new syntactic dataset and perceptual metric that better capture mid-level features such as pose and viewing angle. In addition, the proposed model can learn from human similarity ratings to uncover possible hidden factors that define visual similarity, as shown via extensive evaluations.